# Prefix-Cache-Aware Data Reordering for LLM-Augmented Database Analytics

Yingze Li [1 2]  Dong Wang [1]  Yiming Guo [1]  Yao Chen [3]  Hongzhi Wang [1]  Bingsheng He [2]

## Abstract

LLM-augmented database analytics face a major bottleneck in the costly prefill phase. Although relational tables inherently contain repeated attribute values, standard row-by-row processing produces fragmented prompt layouts that obscure shared prefixes, thereby minimizing opportunities for prefix KV cache reuse and constraining system efficiency. Existing solutions typically employ heuristic or exhaustive search methods to reorder prompt layouts, but these approaches can be inefficient and may not leverage the structural properties of relational tables. We address this challenge by formulating prefix-cache-aware prompt layout optimization as a problem rooted in the isomorphism between prefix-cache reuse and the radix tree topology induced by the relational data distribution. Building on this perspective, we introduce a practical greedy tree-shaping algorithm that efficiently selects row and column orderings to maximize prefix overlap. Our approach, SOLO, improves prefill throughput by up to 90.3% under a fixed prefix-cache budget. Moreover, it reduces planning overhead by up to 242× compared to state-of-the-art baselines.

## 1. Introduction

LLMs are integrated into relational database engines as powerful operators capable of handling complex semantic logic (Databricks, 2025; Microsoft Azure, 2025), thereby enabling a wide range of downstream applications, such as tabular data cleaning (Naeem et al., 2024; Narayan et al., 2022; Li et al., 2024), data integration (Fan et al., 2024; Qiang et al., 2024; Wang et al., 2025b), and semantic

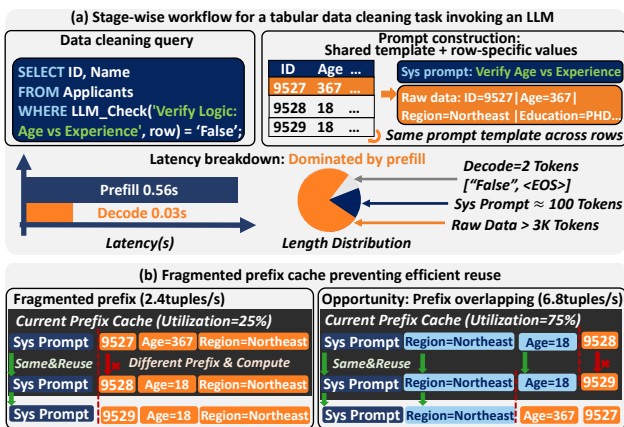

**Figure 1.** Typical LLM-augmented data cleaning in a relational database, where prefill is the major overhead.

data filtering (Lee et al., 2025; Satriani et al., 2025; Wang et al., 2025a), without the need for training task-specific models or maintaining brittle feature pipelines (Konda et al., 2016; Mudgal et al., 2018). In these workloads, the database engine invokes an LLM to process records row-by-row—classifying data quality, extracting entities, or matching descriptions against a query (Liu et al., 2025a; Yuan et al., 2024). As illustrated in Figure 1, in contrast to a conversational AI, a database query triggers an iterative process that feeds a large number of independent rows into the LLM. Each request shares a repetitive prompt template but differs in the specific tuple data (Patel et al., 2025).

Unlike chat-style generation, relational LLM workloads process each tuple individually and typically generate only a small number of output tokens per row, yet repeatedly consume lengthy, schema-shaped prompts containing attribute labels and values (Patel et al., 2025). This leads to workloads that are dominated by the costly prefill phase, where attention Key-Value (KV) states are constructed for each input. For example, in semantic filtering on the Food dataset with Qwen2.5-14B, prefill accounts for over 94.8% of end-to-end latency, far exceeding decoding time (see Figure 1 and Appendix J). **Prefix-cache reuse is limited because the default relational prompt layout, characterized by row-wise request order and column-wise in-prompt attribute order, fails to create shared prefixes between adjacent requests.** Since the prefix cache requires identical prefixes

[1]Faculty of Computing, Harbin Institute of Technology, Harbin, China [2]School of Computing, National University of Singapore, Singapore [3]School of Computer Science and Technology, Huazhong University of Science and Technology, Wuhan, China. Correspondence to: Yao Chen <chenyao_cs@hust.edu.cn>, Hongzhi Wang <wangzh@hit.edu.cn>.

*Proceedings of the 43rd International Conference on Machine Learning*, Seoul, South Korea. PMLR 306, 2026. Copyright 2026 by the author(s).

for reuse, such opportunities are inherently scarce in the default prompt layout. As a result, even though attribute labels are frequently repeated (Lemire & Kaser, 2011; Stonebraker et al., 2005; Lemire et al., 2012), most potential prefix KV reuse is lost without explicitly optimizing request order across rows and serialization order across attributes. Effective cache utilization therefore requires deliberate relational prompt layout optimization to cluster similar prompts and maximize prefix overlap, as illustrated in Figure 1.

Prior work, however, treats prompt layout optimization as a generic scheduling or search problem, relying on heuristics or brute-force recursion that ignore the deep structural dependencies between row and column order; as a result, they suffer from high computational overhead and lack theoretical guarantees (Liu et al., 2025b; Wu et al., 2024; Li & Qiu, 2023; Akella et al., 2024). **We observe that prefix cache reuse in relational workloads is structurally equivalent to path sharing on a radix tree induced by column priority**. Building upon this observation, we formalize that fixing the column order deterministically yields an optimal row ordering via lexicographical sorting, reducing the joint optimization problem to selecting an effective permutation of the $M$ columns. We further show that the objective is prefix-monotone with diminishing returns, enabling a simple and efficient greedy algorithm with strong theoretical guarantees. As a result, our planner offers stable, low-overhead deployment while systematically maximizing prefix-cache reuse and eliminating repeated long-prefix prefills. We instantiate these theoretical findings in our solution Semantic Operator Layout Optimizer (SOLO), a fast and robust prefix-cache-aware scheduler that reorders per-tuple LLM invocation sequence and within-prompt field serialization to maximize prefix overlap between adjacent requests; thereby increasing prefix cache reuse and improving throughput in relational database engines.

Specifically, we make the following contributions:

- We establish a rigorous theoretical framework for prefix-cache-aware relational prompt layout optimization in LLM-augmented relational analytics, revealing an isomorphism between prefix cache reuse and radix tree topology.

- We prove that, for any fixed column order, lexicographical row sorting is optimal for maximizing prefix reuse, and we show the column ordering problem is NP-hard.

- We design an efficient algorithm for column ordering with a tight $1/(M-1)$ worst-case approximation guarantee and provable optimality under the product-uniform independence model, reducing planning overhead to near-linear in data size.

- We implement and integrate our solution, SOLO, into production LLM serving and semantic analytics pipelines,

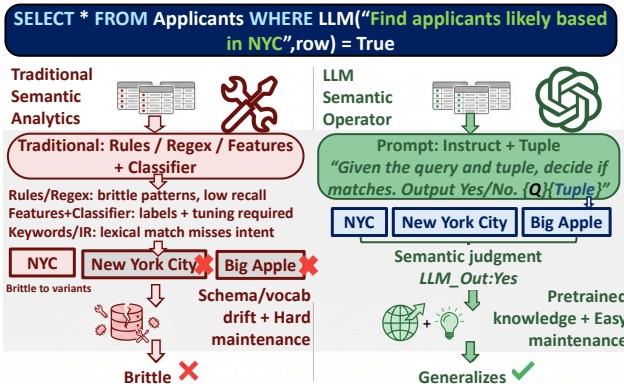

*Figure 2.* Traditional semantic analytics rely on brittle rule/feature pipelines, while LLM-based semantic operators make instruction-guided judgments that are robust to data variants and drift.

demonstrating up to 90.3% throughput improvement and over $242\times$ reduction in planning overhead across real-world datasets and applications.

## 2. Background and Motivation

Large Language Models (LLMs) are rapidly being integrated into databases (DBs) as built-in semantic capabilities, increasingly exposed via SQL functions or operator interfaces in major cloud data platforms (Microsoft Azure, 2025; Snowflake Inc., 2024). This integration addresses the demand for semantic analytics (Konda et al., 2016; Patel et al., 2025)—spanning cleaning, integration, and querying (Naeem et al., 2024; Narayan et al., 2022; Fan et al., 2024; Qiang et al., 2024; Lee et al., 2025; Satriani et al., 2025)—that was previously served by brittle rule-based or task-specific ML pipelines vulnerable to data drift (Dallachiesa et al., 2013; Chu et al., 2015; Doan et al., 2020; Bayardo et al., 2007; Mudgal et al., 2018). Modern systems therefore utilize LLMs as semantic operators to perform natural language instruction-conditioned reasoning directly within the analytical engine (Patel et al., 2025). However, this flexibility introduces a critical bottleneck: row-wise evaluation triggers excessive LLM calls, causing inference costs to dominate end-to-end runtime at scale (Liu et al., 2025a; Yuan et al., 2024).

### 2.1. Efficiency optimizations for semantic operators

A common thread is reorganizing data or requests so repeated structure becomes adjacent during execution, reducing the cost of computation by the managed locality. Traditional DBs create access/compression locality via reordering rows and columns, while LLM serving exploits prefix locality for prefix KV-cache reuse. Semantic operators lie at their intersection: their prompt stream is induced by the operator's logical tuple iteration order and the prompt con-

structor's attribute serialization policy, so locality can be created at the operator/serialization layer and exploited by the serving stack.

**Locality optimizations in relational tables.** Relational tables exhibit repetition and correlations (Stonebraker et al., 2005; Lemire & Kaser, 2011); extensive database research has leveraged data reordering to enhance locality that reduces I/O and improves compressibility (Lemire et al., 2012). Representative examples include columnar storage and compression (Abadi et al., 2008), tuple clustering and locality-aware layout (Binnig et al., 2015; Kang et al., 2021), and entropy-based partitioning and compression-oriented designs (Abadi et al., 2006). The shared takeaway is an adjacency effect: placing similar values near each other increases both access locality and compression effectiveness. We draw inspiration from this principle that reordering rows and columns can systematically increase locality; however, unlike prior physical layout optimization techniques that materialize a new on-disk layout, we apply the same adjacency intuition at prompt scheduling time by logical reordering the relational prompt layout (tuple access order and prompt-field serialization order) without changing the underlying physical table layout.

**Prefix locality optimization via layout-aware semantic operator scheduling.** While LLM serving systems such as PagedAttention (Kwon et al., 2023), FlashAttention (Dao et al., 2022), and SGLang (Sheng et al., 2023) improve KV-cache management and exploit prefix locality when it exists, they treat the request stream as fixed—so fragmented prompt layouts limit their effectiveness. In contrast, semantic operators over relational tables can proactively induce prefix locality by reordering rows and columns, directly impacting prompt similarity and cache reuse (Liu et al., 2025b). Among approaches for reducing the cost of such tabular semantic operators, we focus on the quality-preserving line that restructures execution to increase prefix-cache reuse, rather than trading off quality for speed via smaller models (Patel et al., 2025; Zeighami et al., 2025). Most closely related, OPHR and GGR (Liu et al., 2025b) reorder rows and columns to maximize prefix overlap, reporting up to $3.4\times$ speedup and $32\%$ cost reduction. However, these methods suffer from poor scalability: OPHR is exponential-time, and GGR incurs high planning overhead with no theoretical guarantees, leading to unpredictable cache efficiency on large datasets, complexity details in Appendix G and Table 1.

## 2.2. Motivation: the missing gap

Despite advances in both LLM serving and semantic-operators, a gap remains between prefix-cache reuse mechanisms and the data organization that induces the prompt

*Table 1.* Comparison of existing layout optimization on LLM relational workloads ($M$: number of columns, $N$: number of rows).

| Method | Time Complexity | Guarantee |
|--------|-----------------|-----------|
| OPHR | $\mathcal{O}(N!(M!)^N)$ | Optimal |
| GGR | $\mathcal{O}(M^2N^2(M+N))$ | None |
| **Ours** | $\mathcal{O}(M^2N + MN\log N)$ | $1/(M-1)$ |

stream. LLM serving systems improve attention kernels, cache management, and reuse policies, but generally cannot create prefix locality when prompts are fragmented due to imperfect data layout. Traditional DB semantic operators can create locality by reordering rows/fields, yet existing layout/scheduling methods are computationally expensive and non-scalable, and provide limited theoretical guarantees on achieved reuse. What is missing is a principled framework that connects row-/column-wise repeated structure in relational data to prefix reuse, producing cache-friendly prompt streams with scalable optimization and predictable quality. Our work closes this gap by establishing a structured tree view that maps prefix repetition to path sharing on a radix tree, enabling scalable layout optimization with provable reuse guarantees.

## 3. Problem Formulation and Optimization

In this section, we formulate the locality gap (Section 2) as an optimization problem driven by prefix reuse: given a stream of relational prompts that have repeated attributes, we seek a strategy that maximizes prefix locality between consecutive requests so that the prefix cache can exploit it, thereby minimizing repeated prefill computation.

### 3.1. Problem Formalization

To minimize the cumulative prefill overhead when processing relational data, we leverage prefix caching to reuse calculated prefix Key-Value (KV) states. The reduction in overhead is maximized by optimizing the prompt-stream layout to increase the prefix overlap between consecutive requests. This involves two synergistic dimensions: (i) Structural sharing, where the column permutation $\pi_c$ controls the attribute sequence within prompts to create common prefixes across tuples; and (ii) Processing continuity, where the row permutation $\pi_r$ determines the execution order to ensure that prompts sharing prefixes are processed contiguously within the cache's temporal locality. We thus formulate this task as a joint optimization problem over the search space $\langle \pi_r, \pi_c \rangle$.

Let $R$ be a relation with $N$ tuples $\{t_1, \ldots, t_N\}$ over $M$ attributes $\{A_1, \ldots, A_M\}$. A relational prompt-stream layout is a pair of permutations $\langle \pi_r, \pi_c \rangle$ over rows and columns.

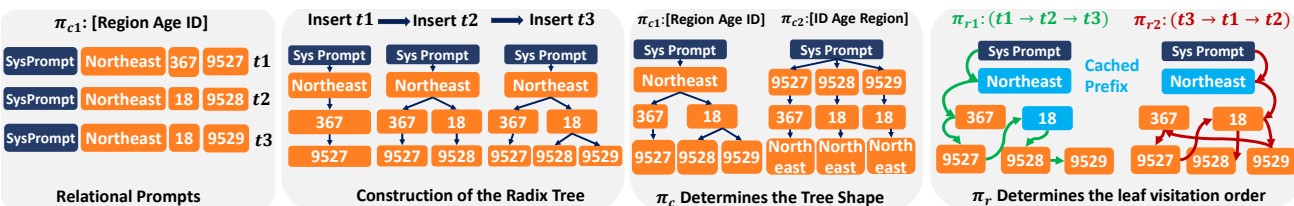

*Figure 3.* From Prefix Overlap to a Radix Tree: $\pi_c$ controls the tree shape while $\pi_r$ determines the leaf visitation order.

Given a fixed query prefix $Q$, the $i$-th prompt is

$$\text{Prompt}_{\pi_r,\pi_c}(i) = \Big\langle Q, t_{\pi_r(i)}[A_{\pi_c(1)}], \ldots, \tag{1}$$
$$t_{\pi_r(i)}[A_{\pi_c(M)}]\Big\rangle.$$

For a fixed column order $\pi_c$, we denote the longest common prefix (LCP) length between two tuples as

$$\kappa(t_i, t_{i'}, \pi_c) = \max\Big\{k \in \{0, \ldots, M\} \mid \forall 1 \le j \le k,$$
$$t_i[A_{\pi_c(j)}] = t_{i'}[A_{\pi_c(j)}]\Big\}. \tag{2}$$

**Assumption 3.1.** When processing $\text{Prompt}_{\pi_r,\pi_c}(i)$, only the prefix states from $\text{Prompt}_{\pi_r,\pi_c}(i-1)$ are guaranteed reusable; thus the effective carry-over equals $\kappa(t_{\pi_r(i-1)}, t_{\pi_r(i)}, \pi_c)$ (Liu et al., 2025b).

Modern prefix-cache implementations, including our vLLM setup, can reuse any still-resident cached prefix. We use Assumption 3.1 as a conservative surrogate because adjacent prefixes are the most reliably resident under finite cache budgets; Section 5.2 empirically validates that adjacent overlap closely predicts realized KV reuse and prefill throughput under global prefix caching.

While $\kappa(\cdot)$ quantifies local reusability between adjacent pairs, optimizing the full workload requires a global structure to capture the aggregate sharing potential across the dataset. Fundamentally, under a fixed column order $\pi_c$, each tuple is linearized into a value sequence, effectively treating the prompt set as a collection of paths starting with the same state $Q$. Prefix caching exploits the resulting topology: where paths share prefixes, their prefix states can be reused; where they diverge, new prefill computation is required. This hierarchical redundancy naturally motivates the use of a Radix Tree, denoted as $\mathcal{T}(R, \pi_c)$. By merging common prefixes into shared internal nodes, the tree provides a global view where the transition cost between any two tuples corresponds to half of the distance between their leaves in the structure.

**Radix-tree interpretation.** Under a fixed column order $\pi_c$, each tuple $t_i$ induces a length-$M$ sequence $\big(t_i[A_{\pi_c(1)}], \ldots, t_i[A_{\pi_c(M)}]\big)$. We construct a radix tree $\mathcal{T}(R, \pi_c)$ by inserting one root-to-leaf path per tuple, where the edge at depth $j$ is labeled by the value $t_i[A_{\pi_c(j)}]$. Shared prefixes across tuples correspond to shared internal nodes, exactly matching the overlap and diverge pattern in Figure 3.

Let $\ell(t_i)$ denote the leaf of $t_i$ in $\mathcal{T}(R, \pi_c)$ and let $\text{d}(\cdot)$ be the node depth (root depth 0). Then for any two tuples $t, t'$, their LCP equals the depth of their Lowest Common Ancestor (LCA) in tree $\mathcal{T}(R, \pi_c)$:

$$\text{d}(\mathcal{LCA}(\ell(t), \ell(t') \mid \pi_c)) = \kappa(t, t', \pi_c). \tag{3}$$

Under this view, the attribute-level prefill cost of transitioning from $t \to t'$ represents the number of new nodes that have to be computed when moving between two leaves. Specifically,

$$\text{cost}_{\pi_c}(t \to t') = M - \text{d}(\mathcal{LCA}(\ell(t), \ell(t') \mid \pi_c)) \tag{4}$$

corresponds to the length of the path from the LCA to the leaf $\ell(t')$. Minimizing the total cost over a sequence of tuples is thus equivalent to finding a traversal of the leaves that minimizes the sum of these "jump" costs.

Thus, considering the first tuple incurs a base cost $M$ to fill the initial path, the total prefill cost $\mathcal{C}(\pi_r, \pi_c)$ over the full sequence is:

$$\mathcal{C}(\pi_r, \pi_c) \triangleq M + \sum_{i=1}^{N-1} \Big( M - \text{d}\big(\mathcal{LCA}(\ell(t_{\pi_r(i)}),$$
$$\ell(t_{\pi_r(i+1)}) \mid \pi_c)\big)\Big). \tag{5}$$

We formalize the objective as:

**Problem 3.1.** *Given a relation $R$, find a row permutation $\pi_r$ and a column permutation $\pi_c$ that minimize the total prefill cost:*

$$(\pi_r^*, \pi_c^*) \in \arg\min_{\pi_r, \pi_c} \mathcal{C}(\pi_r, \pi_c).$$

Equation (5) separates the optimization into two dimensions, as visualized in Figure 3. Structurally, the column order $\pi_c$ controls the tree shape, which determines the global potential for prefix sharing. Sequentially, the row permutation $\pi_r$ governs the leaf traversal order within a fixed tree, thereby controlling how much of that potential is realized under prefix-cache reuse. We leverage this structural decomposition to solve Problem 3.1.

## 3.2. Radix Tree Guided Layout Optimization

We solve Problem 3.1 by separating row and column decisions. In Section 3.2.1, we first prove that for any fixed radix tree induced by a column ordering, the cost-minimizing way is to visit the leaves using a depth-first search (DFS) traversal; moreover, this optimal value under fixed column ordering is the tree size. The remaining task is to choose a column ordering to minimize the induced tree size, for which we prove its intractability and give a bounded approximation algorithm in Section 3.2.2.

### 3.2.1. LEAF TRAVERSAL FOR ROW ORDERING

Our goal is to maximize prefix reuse across consecutive prompts. In the induced radix tree, this requires that once the traversal enters a subtree, it should enumerate all leaves in that subtree before moving to a sibling; otherwise, the traversal discards the current prefix state and later re-prefills it. This requirement naturally leads to the following:

**Theorem 3.2.** *For any fixed column permutation $\pi_c$, the depth-first leaf order $\pi_r^{\mathrm{DFS}}(\pi_c)$ minimizes $\mathcal{C}(\pi_r, \pi_c)$ over all row permutations $\pi_r$, and the minimum cost equals the induced radix tree size $|\mathcal{T}(R, \pi_c)|$:*

$$\min_{\pi_r} \mathcal{C}(\pi_r, \pi_c) = \mathcal{C}\big(\pi_r^{\mathrm{DFS}}(\pi_c), \pi_c\big) = |\mathcal{T}(R, \pi_c)|$$

*Proof sketch.* Interleaving leaves from different subtrees forces the traversal to exit and re-enter a shared prefix state, incurring repeated prefill overhead. By regrouping such interleaved leaves into contiguous blocks, we eliminate these re-entries without decreasing the LCA depth of any transition, thereby strictly minimizing the total cost. This optimality implies a subtree-contiguous depth-first search (DFS). Crucially, under DFS, each non-root node in $\mathcal{T}(R, \pi_c)$ is prefilled exactly once—upon the first entry to its leaf block—and never revisited. Consequently, the total prefill cost equals the number of distinct non-root nodes, $|\mathcal{T}(R, \pi_c)|$. $\square$

Theorem 3.2 implies that, after optimizing rows, minimizing $\mathcal{C}(\pi_r, \pi_c)$ is equivalent to minimizing $|\mathcal{T}(R, \pi_c)|$ over column permutations $\pi_c$. For implementation, this $\pi_r^{\mathrm{DFS}}(\pi_c)$ can be implemented most efficiently by visiting children in lexicographical order of their edge labels: the resulting leaf sequence is exactly the lexicographical order of tuples under $\pi_c$. Thus, lexicographical sorting is an implementation of the same subtree-contiguous DFS, avoiding explicit tree construction.

### 3.2.2. TREE SHAPING FOR COLUMN ORDERING

Theorem 3.2 reduces the problem to minimizing the tree size $|\mathcal{T}(R, \pi_c)|$. However, we find out it is intractable:

**Theorem 3.3.** *Minimizing $|\mathcal{T}(R, \pi_c)|$ over column permutations $\pi_c$ is NP-hard.*

*Proof sketch.* We prove NP-hardness via a polynomial-time reduction from the Minimum Sum Vertex Cover (MSVC) problem which is known to be NP-hard (Bansal et al., 2023). Given an MSVC instance, we construct a relational instance whose optimal column ordering corresponds to an optimal MSVC solution; under this construction, the tree size $|\mathcal{T}(R, \pi_c)|$ equals the MSVC objective. Since MSVC is known to be NP-hard, so is our tree-shaping problem. The complete reduction is detailed in Appendix A. $\square$

As a consequence, unless $P = NP$, no polynomial-time algorithm can guarantee the exact optimum for every instance, which motivates the development of an efficient greedy heuristic with provable approximation guarantees.

To construct this heuristic, we consider growing the radix tree top-down from the root toward the leaves. The first selected column creates depth-one branches; the next refines nodes into children. The key to minimizing the tree size $|\mathcal{T}(R, \pi_c)|$ is controlling how fast the tree branches. Aggressively splitting high-cardinality columns quickly creates many children early on, expanding the tree width and increasing distinct prefix states. Conversely, conservative splits of low-cardinality columns keep the tree compact in upper levels, preserving shared prefixes for longer. Motivated by this, we propose to select the column that minimizes the number of newly created children at each depth.

To derive the theoretical guarantee, we rewrite the objective in a depth-decomposable form. For a fixed column order $\pi_c$, let $V_{\pi_c}^{(k)}$ denote the number of distinct nodes at depth $k$ in the induced radix tree $\mathcal{T}(R, \pi_c)$, which corresponds to the number of distinct length-$k$ prefixes under $\pi_c$. The induced radix tree size defined as the total count of non-root nodes is

$$|\mathcal{T}(R, \pi_c)| \triangleq \sum_{k=1}^{M} V_{\pi_c}^{(k)}. \tag{6}$$

Without reuse, each of the $N$ tuples requires $M$ attribute-level prefills, giving a baseline cost $MN$. We therefore define the equivalent *prefill-savings* objective:

$$\mathcal{H}(\pi_c) \triangleq MN - |\mathcal{T}(R, \pi_c)| = \sum_{k=1}^{M} \left( N - V_{\pi_c}^{(k)} \right), \tag{7}$$

where the second equality follows from Eq. (6). Thus, minimizing $|\mathcal{T}(R, \pi_c)|$ is equivalent to maximizing $\mathcal{H}(\pi_c)$.

This greedy method is robust because the savings objective in Eq. (7) decomposes by depth and exhibits diminishing returns. Recall that $V_{\pi_c}^{(k)}$ counts the number of distinct length-$k$ prefixes, so the per-depth saving is $N - V_{\pi_c}^{(k)}$. Since longer prefixes only refine partitions, we have $V_{\pi_c}^{(1)} \leq \cdots \leq$

$V_{\pi_c}^{(M)} = N$, and thus $N - V_{\pi_c}^{(1)} \geq \cdots \geq N - V_{\pi_c}^{(M)} = 0$. Thus, most savings are front-loaded at shallow depths, and the greedy rule that minimizes the newly created distinct prefixes at each depth, captures a large fraction of the total savings. This yields the following tight worst-case guarantee:

**Theorem 3.4.** *Let $\mathcal{H}_{\mathrm{SOLO}}$ be the objective value achieved by the greedy construction, and $\mathcal{H}_{OPT}$ be the global optimum. Assume $M \geq 2$. Then*

$$\mathcal{H}_{\mathrm{SOLO}} \geq \frac{1}{M-1} \cdot \mathcal{H}_{OPT}.$$

*Proof sketch.* We view each column split as an action whose marginal contribution to $\mathcal{H}$ is its prefix-saving gain. These gains form a sequence with depth-wise diminishing returns: the marginal benefit of applying a split at a deeper level is at most the benefit it would have at the root. The greedy algorithm picks, at each step, the split with the maximum available marginal gain and thus secures the largest possible root-level improvement. Since any optimal solution can use at most $M-1$ distinct column splits along a root-to-leaf path, its total gain is at most $(M-1)$ times the greedy gain at the first step. A rigorous proof is provided in Appendix C.3. $\square$

The worst-case bound in Theorem 3.4 is tight because column correlations couple splitting decisions across depths: an early split changes the conditional distribution inside each prefix group and can alter the marginal gain of later columns. We therefore state the complementary average-case result under the product-uniform model proved in Appendix C.4: tuples are i.i.d., columns are mutually independent, and each column is uniformly distributed over a finite alphabet. In this stylized regime, the coupling disappears: within any prefix group, every remaining column has the same marginal distribution, so the expected immediate gain used by greedy does not hide future penalties. Consequently, the locally optimal choice coincides with the globally optimal ordering in expectation.

**Proposition 3.5.** *Under the product-uniform independence model, let $R$ be generated by drawing $N$ tuples i.i.d. from a product-uniform distribution over attributes. Then the greedy top-down construction maximizes the expected prefill saving objective:*

$$\mathbb{E}[\mathcal{H}(\pi_c^{\mathrm{SOLO}})] = \max_{\pi_c} \mathbb{E}[\mathcal{H}(\pi_c)].$$

*Proof sketch.* By independence, the expected marginal improvement contributed by choosing a column at the next depth is invariant to the already-chosen prefix (it depends only on that column's marginal statistics). Hence maximizing the immediate expected gain at each step maximizes the global expected sum in Eq. (7). Rigorous proof is in Appendix C.4. $\square$

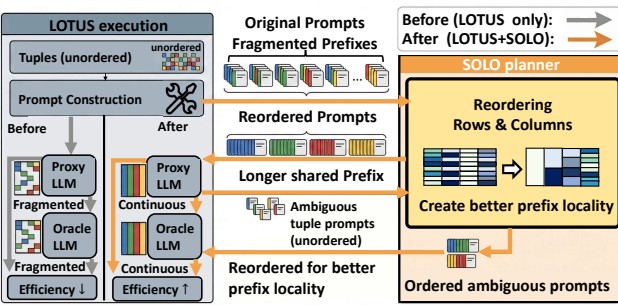

*Figure 4.* Integrate SOLO into the LOTUS semantic filter pipeline.

### 3.3. Computational complexity analysis

The greedy search for the column order $\pi_c$ runs for $M$ rounds and scores all remaining columns per round, yielding $\mathcal{O}(M^2)$ marginal-gain evaluations. Under the radix-tree view, scoring a candidate at depth $k$ amounts to counting the number of distinct length-$k$ branches it would induce. A naive approach computes this count by lexicographically sorting the $N$ tuples by the tentative length-$k$ prefix and then counting groups, which costs $\mathcal{O}(N \log N)$ comparisons; each comparison may inspect up to $k \leq M$ attributes, giving $\mathcal{O}(MN \log N)$ time per evaluation in the worst case. Across $\mathcal{O}(M^2)$ evaluations, the planning time becomes $\mathcal{O}(M^3 N \log N)$. We can avoid repeated sorting by maintaining an incremental prefix state and using hash-based counting, reducing each evaluation to expected $\mathcal{O}(N)$ time and the planning cost to $\mathcal{O}(M^2 N)$, as detailed in Section 4.1. After fixing $\pi_c$, we obtain $\pi_r$ via a single lexicographical sort in $\mathcal{O}(MN \log N)$, so the total expected complexity is $\mathcal{O}(M^2 N + MN \log N)$. The algorithm requires $\mathcal{O}(MN)$ space, dominated by the encoded attribute matrix; empirical scaling is in Appendix D.3.

## 4. Implementation and System Integration

### 4.1. The Optimization Algorithm

This section provides an efficient optimization algorithm for SOLO instantiation. According to Theorem 3.2, once $\pi_c$ is fixed, the optimal $\pi_r$ is the lexicographic order induced by $\pi_c$. Hence, we search only for $\pi_c$ and compute $\pi_r$ via a single final row sort. The main challenge is to evaluate marginal utility for $\pi_c$ without repeatedly row sorting.

Algorithm 1 addresses this challenge by maintaining a rolling prefix state. We integer-encode each column and maintain (i) the current prefix $\mathcal{S}$ which will become $\pi_c$ and (ii) a state vector $F \in \mathbb{Z}^N$ (Lines 3–5), where $F[j]$ is the exact equivalence-class ID of row $j$ induced by $\mathcal{S}$. At depth $k$ (Line 6), to score a candidate next attribute $a \in \mathcal{U}_{\mathrm{rem}}$, we build composite keys $K_{\mathrm{cand}}[j] = (F[j], R_{\mathrm{enc}}[j, a])$ for all rows $j$ (Line 8). The number of distinct keys, $V_{\mathcal{S} \oplus a}^{(k)} = \text{CountUnique}(K_{\mathrm{cand}})$ (Lines 9–11), is exactly

**Algorithm 1** Layout Optimization Algorithm

1: **Input:** Dataset $R$ with $N$ rows; attributes $\mathcal{U} = \{A_1, \ldots, A_M\}$
2: **Output:** Row permutation $\pi_r$; column permutation $\pi_c$
3: $R_{enc} \leftarrow \text{VALUEENCODE}(R)$    // Map values to integers
4: $F \leftarrow \mathbf{0}_N$                                  // Rolling key
5: $\mathcal{S} \leftarrow [\,]; \; \mathcal{U}_{\text{rem}} \leftarrow \mathcal{U}$
6: **for** $k = 1$ to $M$ **do**
7:     $\Delta^\star \leftarrow -\infty, \; a^\star \leftarrow \text{null}, \; K_{\text{best}} \leftarrow \text{null}$
8:     **for** each $a \in \mathcal{U}_{\text{rem}}$ **do**
9:        $K_{\text{cand}} \leftarrow (F, \; R_{enc}[:,a])$    // Candidate tuple keys
10:        $V_{\mathcal{S} \oplus a}^{(k)} \leftarrow \text{COUNTUNIQUE}(K_{\text{cand}})$ // Distinct groups
11:        $\Delta_a \leftarrow N - V_{\mathcal{S} \oplus a}^{(k)}$            // Marginal gain
12:        **if** $\Delta_a > \Delta^\star$ **then**
13:           $\Delta^\star \leftarrow \Delta_a, \; a^\star \leftarrow a, \; K_{\text{best}} \leftarrow K_{\text{cand}}$
14:        **end if**
15:     **end for**
16:     $\mathcal{S} \leftarrow \mathcal{S} \oplus a^\star; \; F \leftarrow \text{FACT}(K_{\text{best}}); \mathcal{U}_{\text{rem}} \leftarrow \mathcal{U}_{\text{rem}} \setminus \{a^\star\}$
17: **end for**
18: $\pi_r \leftarrow \text{LEXSORT}(R, \mathcal{S}); \; \pi_c \leftarrow \mathcal{S}$
19: **return** $\pi_r, \pi_c$

the number of refined prefix groups under $\mathcal{S} \oplus a$, yielding $\Delta_a = N - V_{\mathcal{S} \oplus a}^{(k)}$ (Lines 12–14). Since COUNTUNIQUE is hash-based, each candidate is evaluated in expected $\mathcal{O}(N)$ time without sorting rows. After selecting $a^\star$, we update $\mathcal{S} \leftarrow \mathcal{S} \oplus a^\star$ and set $F \leftarrow \text{FACT}(K_{\text{best}})$ (Line 16), where $\text{FACT}(\cdot)$ factorizes the composite keys by mapping each distinct key to a dense integer ID, keeping $F$ compact across iterations. After $M$ rounds, we perform one lexicographical sort in Line 18 and obtain the full $\pi_r, \pi_c$.

### 4.2. System Integration

We integrate SOLO into the LOTUS system (Patel et al., 2025) as a non-intrusive layout planner, as illustrated in Figure 4. In proxy-based scanning cascades, SOLO operates at batch boundaries: it optimizes the initial stream for the proxy model and subsequently re-orders the filtered set of ambiguous tuples for the Oracle LLM. This two-stage reordering maximizes prefix-cache reuse across the entire cascade without altering the original system design. This optimization strategy generalizes to other operators with commutative intermediate results. Similarly, for Semantic Top-$k$, SOLO accelerates intermediate rounds by reordering batches of pairwise comparisons; for Semantic Join, it optimizes the projection phase by grouping inputs prior to key prediction. Overall, SOLO provides a drop-in acceleration layer that improves end-to-end efficiency via prefix-aware reordering, while preserving the semantics and interfaces of existing systems.

## 5. Experiments

### 5.1. Experimental Setup

**Datasets.** We use four real-world structured datasets: **OpenPayments** (Centers for Medicare and Medicaid Services, 2025), **DMV** (New York State Department of Motor Vehicles, 2019), **Food** (Open Food Facts Association, 2025), **GoogleAnalytics** (Pearse, 2024). For functional-dependency robustness, we additionally use **Flight**, a 110-column airline FD benchmark (Papenbrock & Naumann, 2016). Unless otherwise noted, each semantic scan processes $N_{\text{scan}} = 100\text{K}$ tuples. Appendix D.1 reports prompt-length statistics under the scan template.

**Baselines.** We compare SOLO with the original layout, prior prompt-layout methods, meta-heuristic column search, and handcrafted ordering heuristics. Table 2 defines the layout baselines used in the main and appendix comparisons. The main comparison reports representative families; Appendix D.3 gives the full ranking over all methods.

**Metrics.** Under prefix caching and a fixed KV-cache budget, we report two overall throughput metrics: (i) **Prefill Throughput** (tokens/s), the serving-side prefill token processing rate; and (ii) **Operator Throughput** (tuples/s), the application-level tuples processed per second for a complete semantic operator run. To explain the speedup, we track a reuse chain during execution: (iii) **Average Prefix Overlap Length**; (iv) **Average Reused KV Tokens**; and (v) **Prefix-Cache Hit Rate**.

**Experiment configurations.** All experiments run on a server with $4\times$ NVIDIA RTX A6000 GPUs, 40 Intel Xeon Silver 4210R CPU cores, and 504GB RAM. We use vLLM (v0.10.0) with prefix caching enabled. For downstream evaluations, we integrate SOLO into LOTUS (Patel et al., 2025) and benchmark filter, join, and top-$k$ scoring. Semantic operators use Qwen2.5-14B (Qwen et al., 2024); proxy-based filtering uses Qwen2.5-0.5B for proxy decisions and Qwen2.5-14B for final scoring. All methods share the same tokenizer, prompt template, batching policy, and KV-cache budget. OPHR is an exact exponential-time layout optimizer, so Appendix E reports it in the small-scale exact comparison. Reproducibility details are in Appendix F.

### 5.2. Performance Validation

We evaluate SOLO against three questions: (1) Effectiveness: Does SOLO improve system-level scan efficiency under a fixed cache budget? (2) Efficiency: Is the planning overhead sufficiently lightweight to be amortized by serving-time improvements? (3) Mechanism stability: Does the same prefix-reuse mechanism remain stable under changes in cache behavior, layout components, and correlated schemas?

*Table 2.* Layout baselines.

| Family | Baseline definition |
|---|---|
| Default layout | **Default:** original row and prompt-column order. |
| Prompt-layout methods | **GGR:** Greedy Group Recursion over relational prompts (Liu et al., 2025b).
**OPHR:** exact row-and-column search (Liu et al., 2025b); Appendix E gives the small-scale comparison. |
| Meta-heuristics | **ACLR:** simulated annealing over column order plus lexicographic rows (Bertsimas, 1993).
**GCLR:** genetic search over column order plus lexicographic rows (Mitchell, 1998). |
| Heuristic orders | **Random:** random row order.
**Single-key:** sort rows by one attribute.
**FreqLex:** frequency-based columns plus lexicographic rows.
**Cluster:** group rows by selected prefix keys.
**Vortex:** compression-oriented ordering (Lemire et al., 2012).
**Refgc:** frequency columns plus reflected Gray-code rows (Richards, 1986; Knuth, 1998).
**Zgray:** frequency columns plus Z-order/Gray-code rows (Faloutsos, 1986; Lemire et al., 2012).
**NDV:** ascending distinct-count columns plus lexicographic rows (Lemire & Kaser, 2011). |

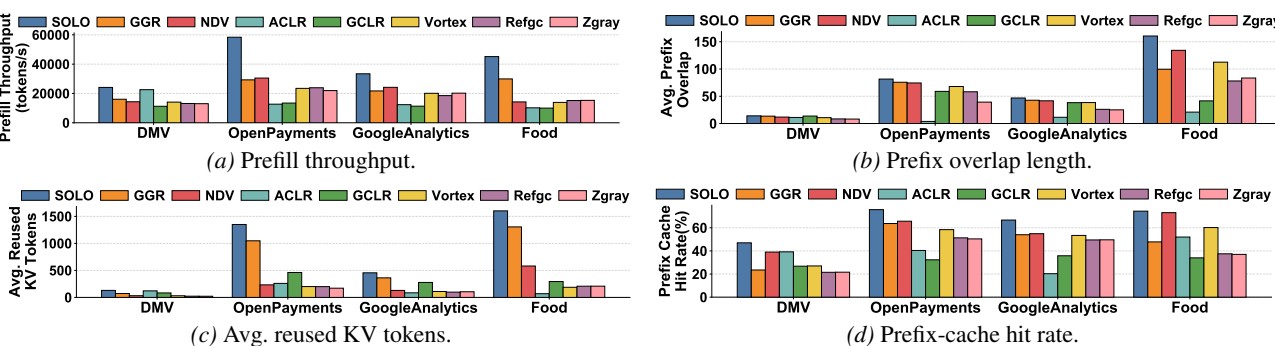

*(a)* Prefill throughput.

*(b)* Prefix overlap length.

*(c)* Avg. reused KV tokens.

*(d)* Prefix-cache hit rate.

*Figure 5.* Serving gains and prefix-cache reuse. SOLO improves prefill throughput; the other panels show the path from adjacent prefix overlap to reused KV tokens and cache hits.

**(1) Effectiveness study.** We first evaluate SOLO as a black-box system optimization under prefix caching and a fixed KV-cache budget. As shown in Figure 5a, SOLO improves prefill throughput by up to 90.3% across datasets, directly demonstrating system-level efficiency gains for semantic scans. We then use the diagnostic metrics in Figure 5 to explain the source of the speedup: SOLO increases adjacent prefix overlap in Figure 5b, which is converted by the serving engine into deeper reused KV tokens in Figure 5c. This higher realized reuse yields a higher prefix-cache hit rate in Figure 5d, which ultimately translates into higher prefill throughput in Figure 5a. Table 3 extends the same effectiveness comparison to the default layout and simple reordering baselines; SOLO has the highest overlap on all four datasets and improves default hit rate by 33.4–71.1 percentage points.

**(2) Efficiency study.** We assess the practicality of reuse planning by measuring its scheduling cost. Figure 6 shows that SOLO is lightweight, reducing planning overhead by up to 242.81× compared to GGR and 1.04×, 13×, 389.56×, 2.32×, 0.96×, and 2.94× compared to NDV, ACLR, GCLR, Vortex, Refgc, and Zgray, respectively. Panel A of Fig-

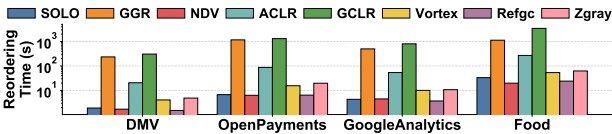

*Figure 6.* Planning overhead.

ure 7 addresses the matched-budget concern: on OpenPayments, SOLO reaches 75.93 prefix overlap in 1.5s, while the strongest compared planner reaches 62.72 even with a 187.5s budget. This shows that SOLO finds a higher-reuse layout faster than existing planners.

**(3) Mechanism stability.** The last question is whether the mechanism behind Figure 5 remains stable: higher prefix overlap should translate into more reused KV tokens and higher prefill throughput. Figure 8 tests the source of this overlap; column ordering and row ordering both contribute, and the full layout gives the largest overlap. Panel B of Figure 7 tests the cache realization step; adjacent overlap is predictive of reused KV tokens and prefill throughput, with Pearson $r \geq 0.993$ and $r \geq 0.981$, respectively. Ta-

*Table 3.* Baseline performance. Overlap is the average number of shared leading attributes; hit rate is the realized prefix-cache hit rate.

| Dataset | #Cols | Default overlap | Default hit rate | Random | Single key | Cluster | FreqLex | NDV | GGR | SOLO overlap | SOLO hit rate |
|---|---|---|---|---|---|---|---|---|---|---|---|
| DMV | 20 | 0.80 | 1.8% | 0.54 | 1.83 | 3.00 | 11.16 | 11.16 | 11.65 | **12.72** | **35.2%** |
| OpenPayments | 90 | 10.49 | 23.7% | 3.36 | 11.32 | 15.94 | 72.48 | 72.47 | 62.72 | **75.93** | **76.3%** |
| GoogleAnalytics | 54 | 1.16 | 3.2% | 0.37 | 2.16 | 4.06 | 40.68 | 40.68 | 39.51 | **44.68** | **62.1%** |
| Food | 200 | 0.00 | 0.0% | 0.00 | 1.00 | 2.00 | 140.43 | 140.30 | 99.58 | **154.91** | **71.1%** |

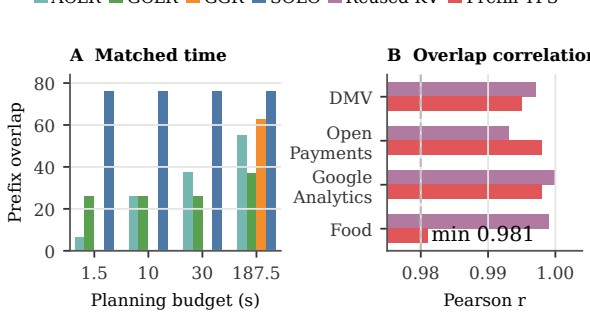

*Figure 7.* Planning budget match (A) and correlation between adjacent KV-token overlap reuse and prefill throughput (B)

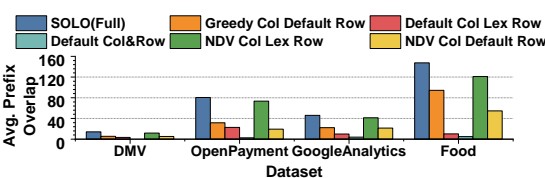

*Figure 8.* Column and row ablation.

ble 4 tests the data side under functional dependencies; at the strongest injected level on OpenPayments, SOLO keeps 65.55 overlap versus 45.26 for NDV. Appendix D.3 reports additional experiments about cache-budget, batch-size, prompt-prefix, and tokenizer.

*Table 4.* Functional-dependency robustness. Synthetic levels denote constrained-column fractions; values are prefix overlap.

| Data | Dependency level | Default | NDV | SOLO |
|---|---|---|---|---|
| Flight | natural | 3.03 | 57.48 | **73.06** |
| OpenPayments | 0% | 10.49 | 72.47 | **75.93** |
| OpenPayments | 20% | 14.10 | 52.47 | **73.75** |
| OpenPayments | 50% | 14.26 | 47.38 | **73.03** |
| OpenPayments | 80% | 26.58 | 45.26 | **65.55** |

### 5.3. Value for Downstream Semantic Operators

For downstream semantic operators, a practical layout optimizer should add little materialization overhead, preserve task outputs, and improve end-to-end operator throughput. At the 100K-tuple scale, SOLO incurs 3.7s on DMV and 17.5s on OpenPayments for planning plus materialization,

while reducing inference time by 26 minutes and 6.0 hours; Appendix D.3 gives planner memory and scaling details. Table 5 shows that reordering preserves output quality: $\Delta$ACC, the accuracy change from the original layout, stays within 0.6 points and unchanged correctness is at least 97.0%. Figure 9 shows that integrating SOLO improves end-to-end throughput for filter, join, and top-$k$ workloads across the four datasets, so the prefix-reuse gains carry from tuple-wise scans to complete semantic operators.

*Table 5.* Quality preservation.

| Dataset | Column-only | | Full layout | |
|---|---|---|---|---|
| | $\Delta$ACC | Unchanged | $\Delta$ACC | Unchanged |
| DMV | +0.4 | 97.8 | +0.4 | 97.6 |
| OpenPayments | -0.4 | 98.8 | -0.2 | 98.6 |
| GoogleAnalytics | -0.6 | 99.4 | -0.6 | 99.4 |
| Food | -0.2 | 97.0 | -0.1 | 97.4 |

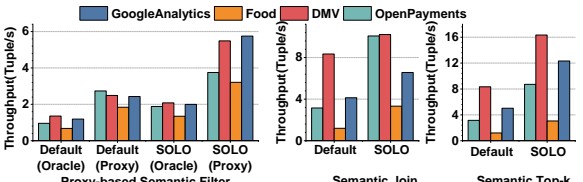

*Figure 9.* End-to-end operator throughput after integrating SOLO into proxy-based semantic filtering, joins, and top-$k$ operators.

## 6. Conclusion

In this paper, we presented **SOLO**, a principled solution for optimizing LLM inference over semantic relational data analytics. By establishing a rigorous isomorphism between prefix KV-cache reuse and radix tree topology, **SOLO** shifts layout optimization from heuristic data shuffling to a well-defined tree shaping problem, where a greedy strategy achieves a tight $1/(M-1)$ approximation guarantee and average-case optimality. SOLO improves prefill throughput by up to 90.3% under a fixed cache budget and reduces planning overhead by up to 242× compared to state-of-the-art baselines. Future work includes hybrid DP-greedy ordering for wide schemas and sketch-based counting, e.g., HyperLogLog++, for further iteration cost reduction.

## Impact Statement

The goal of this work is to improve the system performance of machine learning in relational databases. The proposed method focuses on data layout optimization to enhance hardware utilization. We believe this work primarily impacts the efficiency of ML infrastructure and do not anticipate unique ethical or societal concerns that require specific highlighting.

## Acknowledgments

This work was supported by the National Key Research and Development Program of China (Grant No. 2024YFB4505202) and the Hubei Provincial Natural Science Foundation of China (No. 2026AFA002).

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

## Appendix Roadmap

This appendix collects proofs, algorithmic analyses, and extended experiments that support the main text. Table 6 provides a section-level guide, indicating where to find each theoretical result, complexity bound, and additional empirical evidence.

*Table 6.* Roadmap of Appendices

| Section | Focus | Key Contents |
|---|---|---|
| Appendix A | **NP-hardness** | Polynomial-time reduction from MSVC to Problem 3.1, establishing Theorem 3.3. |
| Appendix B | **Row-order optimality** | Radix tree characterization and proof that DFS/LexSort achieves the tree-size lower bound (Theorem 3.2). |
| Appendix C | **Greedy analysis** | Partition-based notation, monotonicity/diminishing returns, and the tight worst-case $\frac{1}{M-1}$ approximation bound (Section 3.2.2). |
| Appendix D | **Extended experiments** | Dataset and prompt-length statistics, metric decomposition from overlap to realized prefill savings, and additional system slices. |
| Appendix E | **OPHR comparison** | Small-scale comparison to OPHR: overlap gains when feasible and rapid runtime blow-up with timeouts beyond tiny instances. |
| Appendix F | **Reproducibility details** | Serving stack, KV cache budgeting, batching policy, models/decoding, baseline hyperparameters, and hardware configuration. |
| Appendix G | **Baseline complexity** | Worst-case runtime derivation for the GGR heuristic (Theorem G.3). |
| Appendix H | **Case study: entity matching** | Table-structured prompting for EM workloads: throughput/TTFT/reuse/quality metrics under attribute reordering. |
| Appendix I | **Quality stability** | Controlled study of prediction stability under row/column/combined reorderings with paired outcome breakdowns. |
| Appendix J | **Latency Analysis** | Prefill-dominated latency breakdown of semantic operators. |

## A. Detailed Proof of Theorem 3.3

*Proof.* We give a polynomial-time reduction from MSVC.

$g : V \to \{1, \ldots, |V|\}$ minimizing $\sum_{(u,v) \in E} \min(g(u), g(v))$.

**Problem A.1** (Min Sum Vertex Cover (MSVC)). *Given a graph $G = (V, E)$, find an ordering $g : V \to \{1, \ldots, n\}$ that minimizes:*

$$\sum_{(u,v) \in E} \min(g(u), g(v))$$

Construction. Fix $X = |V|^4$. We construct an instance of Problem 3.1 as follows. The relation $\mathcal{R}$ has $M = |V|$ attributes $\{A_v\}_{v \in V}$ and

$$N = |E| \cdot X^2$$

tuples. For each edge $e = (u, v) \in E$, we create an edge gadget $T_e$ consisting of $X^2$ tuples $\{t_{ij}\}_{1 \le i,j \le X}$ defined by

$$t_{ij}[A_k] = \begin{cases} \text{`e'} + (i-1), & k = u, \\ \text{`e'} + (j-1), & k = v, \\ \text{`e'}, & \text{otherwise,} \end{cases} \tag{8}$$

where `e` is a unique identifier string for the edge $e$. Let $\mathcal{R} = \bigcup_{e \in E} T_e$.

The size of $\mathcal{R}$ is polynomial in $|V| + |E|$: it contains $|E| \cdot X^2 = |E| \cdot |V|^8$ tuples, each with $|V|$ attributes, hence writing down all entries takes

$$\mathcal{O}\big(|E| \cdot X^2 \cdot |V|\big) = \mathcal{O}\big(|E| \cdot |V|^9\big)$$

time, which is polynomial.

From columns to a vertex ordering. Fix any column permutation $\pi_c$. Let

$$f(v) \; := \; \text{the position of attribute } A_v \text{ in } \pi_c, \qquad f : V \to \{1, \ldots, |V|\}. \tag{9}$$

By Theorem 3.2, the optimal row permutation under $\pi_c$ is the lexicographical order induced by $\pi_c$. Therefore $\mathcal{H}(\pi_c)$ is fully determined by $\pi_c$ (equivalently by $f$).

Gadgets do not interleave. We first show that, under any $\pi_c$, tuples from different edge gadgets occupy disjoint contiguous blocks in the lexicographically sorted order.

**Lemma A.1.** *For any two distinct edges $e \neq e'$ and any tuples $t \in T_e$, $t' \in T_{e'}$, we have $t[A_k] \neq t'[A_k]$ for every attribute $A_k$. Consequently, under lexicographical sorting by any $\pi_c$, all tuples in $T_e$ appear contiguously, and $\mathcal{H}(\pi_c)$ decomposes additively over edge gadgets:*

$$\mathcal{H}(\pi_c) = \sum_{(u,v) \in E} S_{uv}, \tag{10}$$

*where $S_{uv}$ is the total common-prefix contribution from the block $T_{(u,v)}$.*

*Proof.* By construction, for any $k \in V$, the value $t[A_k]$ is either `e` or `e` $+ \delta$ for some $\delta \in \{0, \dots, X-1\}$, and similarly $t'[A_k]$ is either `e`' or `e`' $+ \delta'$. Choosing identifiers `e` as distinct base strings (or equivalently distinct integer ranges) ensures $\{$ `e` $+ \delta : \delta \in [0, X-1]\}$ and $\{$ `e`' $+ \delta' : \delta' \in [0, X-1]\}$ are disjoint, hence $t[A_k] \neq t'[A_k]$ for all $k$. Therefore, in lexicographical order, any comparison between a tuple in $T_e$ and a tuple in $T_{e'}$ is decided at the first attribute encountered, so tuples from $T_e$ cannot be interleaved with tuples from $T_{e'}$; each $T_e$ forms a contiguous block. Since the objective $\mathcal{H}(\pi_c)$ sums common-prefix lengths over adjacent pairs, adjacency never crosses blocks, implying the additive decomposition (10). $\square$

Contribution of one edge gadget. Fix an edge $e = (u, v) \in E$. We compute $S_{uv}$ as a function of $f(u)$ and $f(v)$.

**Lemma A.2.** *Let $e = (u, v) \in E$ and consider the gadget $T_e$ sorted lexicographically under $\pi_c$. Its contribution to (10) is*

$$S_{uv} = (X-1)\big(\min(f(u), f(v)) - 1\big) + X(X-1)\big(\max(f(u), f(v)) - 1\big). \tag{11}$$

*Proof.* Assume w.l.o.g. $f(u) < f(v)$ (the other case is symmetric). In the gadget $T_e$, attributes other than $A_u$ and $A_v$ are constant `e` across all tuples. Hence lexicographical sorting is determined by $A_u$ first, and then by $A_v$ (because $A_u$ appears earlier than $A_v$ in $\pi_c$).

There are $X$ distinct values for $A_u$, namely `e` $+ (i-1)$ for $i = 1, \dots, X$. Therefore, the sorted order partitions $T_e$ into $X$ consecutive groups indexed by $i$, each containing exactly $X$ tuples with the same $A_u$ value and varying $A_v$ values.

Within each fixed-$i$ group, adjacent tuples differ first at attribute $A_v$ (since all earlier attributes in $\pi_c$ match, including $A_u$), so each such adjacent pair has common-prefix length $f(v) - 1$. Each group contributes $(X-1) \cdot (f(v) - 1)$, and there are $X$ groups, giving

$$\underbrace{X(X-1) \cdot (f(v) - 1)}_{\text{within-group adjacencies}}.$$

Across consecutive groups (from $i$ to $i+1$), the boundary adjacent pair differs first at attribute $A_u$, so its common-prefix length is $f(u) - 1$. There are exactly $(X-1)$ such boundaries, giving

$$\underbrace{(X-1) \cdot (f(u) - 1)}_{\text{between-group boundaries}}.$$

Summing both parts yields

$$S_{uv} = (X-1) \cdot (f(u) - 1) + X(X-1) \cdot (f(v) - 1),$$

which is exactly (11) since $f(u) = \min(f(u), f(v))$ and $f(v) = \max(f(u), f(v))$ under $f(u) < f(v)$. $\square$

Objective rewriting. Combining Lemma A.1 and Lemma A.2, we can rewrite $\mathcal{H}(\pi_c)$. Let

$$C_0 := (X-1) + X(X-1).$$

Then from (11) we have

$$\mathcal{H}(\pi_c) = \sum_{(u,v) \in E} \Big[(X-1)\big(\min(f(u), f(v)) - 1\big) + X(X-1)\big(\max(f(u), f(v)) - 1\big)\Big]$$

$$= (X-1) \sum_{(u,v) \in E} \min(f(u), f(v)) + X(X-1) \sum_{(u,v) \in E} \max(f(u), f(v)) - |E|C_0. \tag{12}$$

Using $\min(a, b) + \max(a, b) = a + b$, we obtain

$$\sum_{(u,v)\in E} \min(f(u), f(v)) = \sum_{(u,v)\in E} \big(f(u) + f(v) - \max(f(u), f(v))\big)$$

$$= \sum_{(u,v)\in E} (f(u) + f(v)) - \sum_{(u,v)\in E} \max(f(u), f(v)). \tag{13}$$

Plugging (13) into (12) gives

$$\mathcal{H}(\pi_c) = (X - 1) \sum_{(u,v)\in E} (f(u) + f(v)) + (X - 1)^2 \sum_{(u,v)\in E} \max(f(u), f(v)) - |E|C_0$$

$$= (X - 1)\Bigg[\underbrace{\sum_{(u,v)\in E} (f(u) + f(v))}_{=:S_{\mathrm{deg}}} + (X - 1)\underbrace{\sum_{(u,v)\in E} \max(f(u), f(v))}_{=:S_{\mathrm{max}}}\Bigg] - |E|C_0. \tag{14}$$

Noting that $\sum_{(u,v)\in E}(f(u) + f(v)) = \sum_{v\in V} \deg(v)f(v)$, we recover exactly

$$\Psi(f) = S_{\mathrm{deg}} + (X - 1)S_{\mathrm{max}}, \tag{15}$$

and maximizing $\mathcal{H}(\pi_c)$ is equivalent to maximizing $\Psi(f)$ since $(X - 1) > 0$ and $-|E|C_0$ is constant.

Dominance of $S_{\mathrm{max}}$. We show that for $X = |V|^4$, any maximizer of $\Psi(f)$ must maximize $S_{\mathrm{max}}$.

**Lemma A.3.** *For any ordering $f$, we have $S_{\mathrm{deg}} < |V|^3$. Moreover, for any two orderings $f_1, f_2$, if $S_{\mathrm{max}}(f_2) \geq S_{\mathrm{max}}(f_1)+1$, then $\Phi(f_2) > \Phi(f_1)$. Consequently, every maximizer of $\Psi(f)$ maximizes $S_{\mathrm{max}}$.*

*Proof.* Since $\deg(v) \leq |V| - 1$ and $f(v) \leq |V|$,

$$S_{\mathrm{deg}} = \sum_{v\in V} \deg(v)f(v) \leq \sum_{v\in V}(|V| - 1)|V| = |V|^2(|V| - 1) < |V|^3.$$

For any two orderings $f_1, f_2$,

$$\Phi(f_2) - \Phi(f_1) = \big(S_{\mathrm{deg}}(f_2) - S_{\mathrm{deg}}(f_1)\big) + (X - 1)\big(S_{\mathrm{max}}(f_2) - S_{\mathrm{max}}(f_1)\big).$$

The first difference is bounded below by $-|V|^3$ (by the above bound on $S_{\mathrm{deg}}$), while if $S_{\mathrm{max}}(f_2) \geq S_{\mathrm{max}}(f_1) + 1$, then the second term is at least $(X - 1) \geq |V|^4 - 1$. Thus for $|V| \geq 2$,

$$\Phi(f_2) - \Phi(f_1) \geq -(|V|^3) + (|V|^4 - 1) > 0,$$

so $\Phi(f_2) > \Phi(f_1)$. Therefore any maximizer of $\Phi$ must maximize $S_{\mathrm{max}}$. $\qquad\square$

Equivalence to MSVC. It remains to relate maximizing $S_{\mathrm{max}}$ to MSVC. Define the reverse ordering

$$g(v) := |V| + 1 - f(v), \tag{16}$$

which is a bijection between orderings $f$ and $g$. For any edge $(u, v) \in E$,

$$\max\big(f(u), f(v)\big) = \max\big(|V| + 1 - g(u), |V| + 1 - g(v)\big)$$

$$= (|V| + 1) - \min\big(g(u), g(v)\big). \tag{17}$$

Summing (17) over $E$ yields

$$S_{\mathrm{max}}(f) = \sum_{(u,v)\in E} \max\big(f(u), f(v)\big)$$

$$= \sum_{(u,v)\in E} \Big((|V| + 1) - \min\big(g(u), g(v)\big)\Big)$$

$$= |E|(|V| + 1) - \sum_{(u,v)\in E} \min\big(g(u), g(v)\big). \tag{18}$$

Since $|E|(|V|+1)$ is constant, maximizing $S_{\max}(f)$ is equivalent to minimizing $\sum_{(u,v)\in E} \min(g(u), g(v))$, which is exactly the MSVC objective.

**Conclusion.** Given an MSVC instance $G = (V, E)$, we built in polynomial time an instance of Problem 3.1 such that an optimal layout (equivalently an optimal $\pi_c$) yields an ordering $g$ solving MSVC, and vice versa. Hence Problem 3.1 is NP-hard. $\qquad\square$

## B. Proof of Theorem 3.2

We fix an arbitrary column permutation $\pi_c$ throughout this appendix and prove Theorem 3.2. Let each tuple $t_i$ be represented as an ordered value vector $\mathbf{v}_i = \langle t_i[A_{\pi_c(1)}], \ldots, t_i[A_{\pi_c(M)}]\rangle$. Let $\mathcal{T} = \mathcal{T}(R, \pi_c)$ be the radix tree obtained by inserting all $\mathbf{v}_i$: each node at depth $k$ corresponds to a distinct length-$k$ prefix, and each tuple corresponds to a root-to-leaf path of length $M$. We write $|\mathrm{Nodes}(\mathcal{T})|$ for the number of *non-root* nodes in $\mathcal{T}$.

**Notation alignment with Section 3.** In the main text, the total attribute-level *prefill* cost is denoted by $\mathcal{C}(\pi_r, \pi_c)$ (Equation 5). Throughout this appendix we write $\mathrm{Cost}(\pi_r \mid \pi_c)$; these two notations coincide:

$$\mathrm{Cost}(\pi_r \mid \pi_c) = \mathcal{C}(\pi_r, \pi_c).$$

Moreover, the induced radix tree size $|\mathcal{T}(R, \pi_c)|$ (Equation 6) equals the number of non-root nodes, hence

$$|\mathcal{T}(R, \pi_c)| = |\mathrm{Nodes}(\mathcal{T})|.$$

Finally, we use $\mathrm{depth}(\cdot)$ interchangeably with $\mathrm{d}(\cdot)$ for node depth (root depth 0), and we let $\ell(t_i)$ denote the leaf corresponding to tuple $t_i$. For two leaves $u, v$, we write $\mathrm{LCA}(u, v \mid \pi_c)$ for their lowest common ancestor in $\mathcal{T}(R, \pi_c)$.

### B.1. Cost under single-step prefix caching

Under Assumption 3.1, when processing a row order $\pi_r$, the reusable prefix between consecutive processed tuples equals the LCP length $\kappa(\cdot)$ under $\pi_c$. Thus, the total number of (attribute-level) computations equals

$$\mathrm{Cost}(\pi_r \mid \pi_c) = M + \sum_{i=1}^{N-1} \left( M - \kappa(t_{\pi_r(i)}, t_{\pi_r(i+1)}, \pi_c) \right), \tag{19}$$

**Connection to the main-text *prefill* objective.** Define the transition cost (Equation 4 in the main text) by

$$\mathrm{cost}_{\pi_c}(t \to t') \; := \; M - \kappa(t, t', \pi_c).$$

Then Equation (19) is exactly the main-text total *prefill* cost $\mathcal{C}(\pi_r, \pi_c)$ in Equation 5, with the first tuple incurring cost $M$.

For two tuples $t_x, t_y$, let $\mathrm{lca}(x, y)$ denote the lowest common ancestor of their corresponding leaves in $\mathcal{T}$. By construction of $\mathcal{T}$, we have

$$\kappa(t_x, t_y, \pi_c) = \mathrm{depth}\big(\mathrm{lca}(x, y)\big). \tag{20}$$

Equivalently, under the main-text notation, letting $u = \ell(t_x)$ and $v = \ell(t_y)$ be the corresponding leaves, we can write

$$\kappa(t_x, t_y, \pi_c) = \mathrm{d}(\mathrm{LCA}(u, v \mid \pi_c)),$$

i.e., the reusable prefix length equals the shared root-to-LCA path length.

### B.2. Lower bound: any schedule must pay at least the tree size

**Lemma B.1** (Tree-size lower bound). *For any row order $\pi_r$,*

$$\mathrm{Cost}(\pi_r \mid \pi_c) \; \geq \; |\mathrm{Nodes}(\mathcal{T})|. \tag{21}$$

*Proof.* Fix any non-root node $u$ in $\mathcal{T}$, and let $\mathcal{L}(u)$ be the set of leaves (tuples) in the subtree rooted at $u$. Consider the *first time* the schedule $\pi_r$ outputs a tuple whose leaf lies in $\mathcal{L}(u)$; let this occur at position $p$ in the output sequence.

If $p = 1$, then the first prompt computes the entire root-to-leaf path, hence it certainly computes the prefix-extension represented by $u$. If $p > 1$, then the previous output tuple at position $p - 1$ is *not* in $\mathcal{L}(u)$ by definition of $p$. Therefore, the LCP between the $(p - 1)$-st and $p$-th tuples has length strictly smaller than $\mathrm{depth}(u)$, i.e., the reusable prefix cannot include node $u$. Under single-step caching, the prompt at position $p$ must compute all attributes beyond the reusable prefix, which includes the prefix-extension represented by $u$. Hence, for each node $u$, we can charge at least one unit of computation to the moment the schedule *first enters* $\mathcal{L}(u)$.

These charges are injective: each node is charged exactly once (at the first entry of its subtree). Thus, the total computation is at least the number of non-root nodes in $\mathcal{T}$. $\qquad\square$

### B.3. Achievability: DFS/lexicographical orders meet the lower bound

We next show that a depth-first traversal of $\mathcal{T}$ produces a leaf order that achieves the lower bound in Lemma B.1. Formally, call a leaf order *DFS-consistent* if, for every internal node $u$, all leaves in $\mathcal{L}(u)$ appear as a contiguous block in the order (equivalently, the order can be realized by a DFS that fully exhausts a subtree before moving to the next).

**Lemma B.2** (DFS achieves tree-size cost)**.** *Let $\pi_r$ be any DFS-consistent leaf order of $\mathcal{T}$. Then*

$$Cost(\pi_r \mid \pi_c) = |\mathrm{Nodes}(\mathcal{T})|. \tag{22}$$

*Proof.* We prove that under a DFS-consistent order, each non-root node in $\mathcal{T}$ is computed *exactly once*. Consider executing the prompts in the order $\pi_r$.

**(i) Each non-root node is computed at least once.** This follows from Lemma B.1 (or directly because every node lies on the path to some leaf).

**(ii) No non-root node is computed more than once.** Fix any node $u$ at depth $d \geq 1$ with parent $p(u)$ at depth $d - 1$. Let $\mathcal{L}(u)$ be its leaf set. Because the order is DFS-consistent, all leaves in $\mathcal{L}(u)$ appear contiguously. Let that contiguous block start at position $s$ and end at position $e$ in the output sequence.

At the moment the schedule enters the block (from position $s - 1$ to $s$, or $s = 1$), the computation must reach depth $d$ along the path into $u$; thus the prefix-extension represented by $u$ is computed at least once at entry.

For any transition *within* the block (between positions $s, \ldots, e$), both leaves lie in the subtree of $u$, so their LCA has depth at least $d$; by Eq. (20), the reusable prefix length is at least $d$. Therefore, node $u$ is always contained in the reusable prefix for all within-block transitions and is never recomputed there.

After leaving the block (from position $e$ to $e + 1$), the schedule never returns to $\mathcal{L}(u)$ again by contiguity, so node $u$ is never recomputed in the future.

Hence $u$ is computed exactly once. Since this holds for every non-root node, the total cost equals $|\mathrm{Nodes}(\mathcal{T})|$. $\qquad\square$

### B.4. Optimality characterization: any optimal order must be DFS-consistent

**Lemma B.3** (Interleaving forces extra cost)**.** *If a leaf order $\pi_r$ is not DFS-consistent, then $Cost(\pi_r \mid \pi_c) > |\mathrm{Nodes}(\mathcal{T})|$.*

*Proof.* If $\pi_r$ is not DFS-consistent, there exists an internal node $u$ such that leaves in two distinct child subtrees of $u$ are interleaved in the order. Let $c_1$ and $c_2$ be two children of $u$, and consider the earliest position where the order switches from a leaf in $\mathcal{L}(c_1)$ to a leaf in $\mathcal{L}(c_2)$, and later switches back to $\mathcal{L}(c_1)$ again. That is, there exist indices $i < j < k$ such that the $i$-th leaf is in $\mathcal{L}(c_1)$, the $j$-th leaf is in $\mathcal{L}(c_2)$, and the $k$-th leaf is in $\mathcal{L}(c_1)$.

Consider the transition into $\mathcal{L}(c_2)$ at step $j$. Since the previous leaf is in $\mathcal{L}(c_1)$, the LCA between these two leaves is exactly $u$ (or lies above $u$), so the reusable prefix length is at most $\mathrm{depth}(u)$. In particular, the edge/node corresponding to child $c_2$ at depth $\mathrm{depth}(u) + 1$ is *not* reusable and must be computed.

Now consider the later transition back into $\mathcal{L}(c_1)$ at step $k$. By the same argument, the node corresponding to child $c_1$ at depth $\mathrm{depth}(u) + 1$ must be computed again upon re-entry.

Thus, at least one child node at depth $\mathrm{depth}(u) + 1$ is computed more than once due to interleaving. Since $|\mathrm{Nodes}(\mathcal{T})|$ counts each non-root node only once, the total cost must strictly exceed $|\mathrm{Nodes}(\mathcal{T})|$. $\qquad\square$

### B.5. Putting together: proof of Theorem 3.2

*Proof of Theorem 3.2.* Fix $\pi_c$ and let $\mathcal{T} = \mathcal{T}(R, \pi_c)$.

**(1) Minimum cost equals tree size.** Lemma B.1 gives $\mathrm{Cost}(\pi_r \mid \pi_c) \geq |\mathrm{Nodes}(\mathcal{T})|$ for any $\pi_r$. Lemma B.2 shows that any DFS-consistent order achieves equality. Hence, the minimum achievable cost is exactly $|\mathrm{Nodes}(\mathcal{T})|$.

**(2) Characterization of optimal row orders.** By Lemma B.3, any non-DFS-consistent order has strictly larger cost than $|\mathrm{Nodes}(\mathcal{T})|$, therefore it cannot be optimal. Thus, $\pi_r$ is optimal if and only if it is DFS-consistent (i.e., corresponds to a DFS leaf order).

**(3) Equivalence to lexicographical sorting (up to ties).** Sorting tuples by the lexicographical key $\langle t[A_{\pi_c(1)}], \ldots, t[A_{\pi_c(M)}]\rangle$ produces an order in which all tuples sharing any fixed prefix form a contiguous block; hence it is DFS-consistent. Conversely, any DFS traversal order can be realized as a nondecreasing lexicographical order by an appropriate tie-breaking among equal keys. Therefore, any nondecreasing lexicographical order (with arbitrary tie-breaking) is optimal, completing the proof. $\qquad\square$

## C. Proofs for Section 3.2.2

### C.1. Radix tree partitions and notation

Let the table contain $N$ tuples (rows) and $M$ columns (attributes). For any ordered column sequence (permutation) $\pi_c = (\pi_1, \ldots, \pi_M)$, the prefix $\pi_{[1:k]}$ induces a partition of tuples into equivalence classes by identical values on the first $k$ columns: two tuples $t, t'$ are equivalent at depth $k$ iff $t[\pi_1] = t'[\pi_1], \ldots, t[\pi_k] = t'[\pi_k]$. Let $P_{\pi_c}^{(k)}$ denote this partition and

$$V_{\pi_c}^{(k)} := |P_{\pi_c}^{(k)}|$$

be the number of distinct length-$k$ prefixes (i.e., the number of nodes at depth $k$ in the induced radix tree).

Recall the reuse objective (Eq. 7 in the main text):

$$\mathcal{H}(\pi_c) \triangleq \sum_{k=1}^{M} \left(N - V_{\pi_c}^{(k)}\right). \tag{23}$$

For convenience, define the per-depth *savings*

$$s_k(\pi_c) := N - V_{\pi_c}^{(k)}, \quad \text{so that} \quad \mathcal{H}(\pi_c) = \sum_{k=1}^{M} s_k(\pi_c).$$

**Greedy construction.** Algorithm 1 builds $\pi_c$ left-to-right. At step $k$ given prefix $\pi_{[1:k-1]}$, it selects the column $a$ maximizing the marginal gain

$$\Delta_k(a \mid \pi_{[1:k-1]}) := \mathcal{H}(\pi_{[1:k-1]} \oplus a) - \mathcal{H}(\pi_{[1:k-1]}) = N - V_{\pi_{[1:k-1]} \oplus a}^{(k)} = s_k(\pi_{[1:k-1]} \oplus a),$$

i.e., it chooses the column that creates the fewest distinct prefixes at the next depth.

### C.2. Monotone improvement and diminishing returns

**Lemma C.1** (Partition refinement along depth). *For any $\pi_c$ and any $k \in \{1, \ldots, M-1\}$, the depth-$(k+1)$ partition refines the depth-$k$ partition: $P_{\pi_c}^{(k+1)}$ is a refinement of $P_{\pi_c}^{(k)}$. Consequently,*

$$V_{\pi_c}^{(1)} \leq V_{\pi_c}^{(2)} \leq \cdots \leq V_{\pi_c}^{(M)}.$$

*Proof.* If two tuples agree on the first $k+1$ columns of $\pi_c$, they also agree on the first $k$ columns. Hence each equivalence class at depth $k+1$ is contained in some equivalence class at depth $k$, which is exactly the refinement relation. Refinement cannot decrease the number of blocks, so $V_{\pi_c}^{(k)} \leq V_{\pi_c}^{(k+1)}$ for all $k$. $\qquad\square$

**Corollary C.2** (Monotone improvement and diminishing returns (per-depth savings))**.** *For any $\pi_c$, the per-depth savings satisfy*

$$s_1(\pi_c) \geq s_2(\pi_c) \geq \cdots \geq s_M(\pi_c) \geq 0.$$

*Proof.* By Lemma C.1, $V_{\pi_c}^{(k)}$ is nondecreasing in $k$, hence $s_k(\pi_c) = N - V_{\pi_c}^{(k)}$ is nonincreasing and nonnegative. $\quad\square$

### C.3. Worst-case approximation guarantee (tight)

We now give a worst-case guarantee for Algorithm 1 under the objective (23).

**Theorem C.3** (Worst-case approximation guarantee (tight))**.** *Assume $M \geq 2$. Let $\pi_c^{\mathrm{SOLO}}$ be the column order returned by Algorithm 1, and $\pi_c^\star$ be an optimal permutation maximizing $\mathcal{H}(\pi_c)$. Then*

$$\mathcal{H}(\pi_c^{\mathrm{SOLO}}) \;\geq\; \frac{1}{M-1}\, \mathcal{H}(\pi_c^\star).$$

*Moreover, the factor $1/(M-1)$ is tight in the worst case.*

*Proof.* Let $s_k(\pi) = N - V_\pi^{(k)}$. By Corollary C.2, for any permutation $\pi$ we have

$$\sum_{k=1}^{M-1} s_k(\pi) \;\leq\; (M-1)\, s_1(\pi), \quad \text{and} \quad \mathcal{H}(\pi) = \sum_{k=1}^{M-1} s_k(\pi) + s_M(\pi).$$

Note that $V_\pi^{(M)}$ counts the number of distinct full tuples (rows) and therefore is *order-invariant*; hence $s_M(\pi)$ is the same for all permutations. Denote this constant by $s_M^{\mathrm{all}}$.

Let $s_1^{\max} := \max_a \big(N - V_{[a]}^{(1)}\big)$ be the best achievable first-depth savings. Since Algorithm 1 maximizes $\Delta_1(a \mid \emptyset) = N - V_{[a]}^{(1)}$, its first choice achieves $s_1^{\max}$. Therefore,

$$\mathcal{H}(\pi_c^{\mathrm{SOLO}}) \;\geq\; s_1^{\max} + s_M^{\mathrm{all}}.$$

For the optimum, using $\sum_{k=1}^{M-1} s_k(\pi^\star) \leq (M-1)s_1(\pi^\star) \leq (M-1)s_1^{\max}$, we have

$$\mathcal{H}(\pi_c^\star) \;=\; \sum_{k=1}^{M-1} s_k(\pi_c^\star) + s_M^{\mathrm{all}} \;\leq\; (M-1)s_1^{\max} + s_M^{\mathrm{all}} \;\leq\; (M-1)\big(s_1^{\max} + s_M^{\mathrm{all}}\big).$$

Combining the two displays yields

$$\mathcal{H}(\pi_c^{\mathrm{SOLO}}) \;\geq\; \frac{1}{M-1}\, \mathcal{H}(\pi_c^\star).$$

$\quad\square$

**Tightness.** We provide an explicit family where the ratio equals $1/(M-1)$. Let $N = m^2$ tuples indexed by $(p, q) \in [m] \times [m]$. Create one column $A := p$ and $(M-1)$ columns $B_1, \ldots, B_{M-1}$ all equal to $q$ (identical as data columns). Then $V_{[A]}^{(1)} = V_{[B_i]}^{(1)} = m$, so the first-step gain ties. If greedy tie-breaking selects $A$ first, then at depth 2 adding any $B_i$ yields $V^{(2)} = N$ (all pairs $(p, q)$), so $s_2 = \cdots = s_M = 0$ and $\mathcal{H}(\pi_c^{\mathrm{SOLO}}) = N - m$. In contrast, the ordering $(B_1, \ldots, B_{M-1}, A)$ satisfies $V^{(k)} = m$ for all $k \leq M-1$ and $V^{(M)} = N$, hence $\mathcal{H}(\pi_c^\star) = (M-1)(N - m)$, giving the ratio exactly $1/(M-1)$.

*Remark* C.4. For $M \geq 2$, the bound $\mathcal{H}(\pi_c^{\mathrm{SOLO}}) \geq \frac{1}{M}\mathcal{H}(\pi_c^\star)$ also follows immediately from Corollary C.2, but it is not tight for permutations over all $M$ columns.

### C.4. Optimality under an independence model

We also analyze a stylized regime where the data distribution factorizes across columns. In this setting, greedy coincides with the globally optimal column order (in expectation).

*Table 7.* Dataset sizes and prompt lengths.

| Dataset | #Rows | #Cols | Input Prompt Token Length | | | | |
|---|---|---|---|---|---|---|---|
| | | | Avg | Min | 50% | 90% | Max |
| OpenPayments | 100000 | 90 | 1478.58 | 1435 | 1478 | 1496 | 1546 |
| DMV | 100000 | 20 | 225.97 | 210 | 226 | 231 | 240 |
| Food | 100000 | 200 | 3120.92 | 2798 | 3065 | 3421 | 7462 |
| GoogleAnalytics | 100000 | 54 | 591.89 | 579 | 590 | 596 | 699 |

**Model.** Assume tuples are i.i.d. draws from a *product* distribution over columns. For each column $a$, values are uniformly distributed over an alphabet of size $K_a$, and columns are mutually independent. Equivalently, for any set of columns $S$, the joint takes values uniformly over $\prod_{a \in S}[K_a]$.

For any $K \in \mathbb{N}$, define the expected number of occupied bins when throwing $N$ balls i.i.d. into $K$ bins uniformly:

$$g(N, K) := K\left(1 - (1 - \tfrac{1}{K})^N\right). \tag{24}$$

**Lemma C.5** (Expected number of distinct prefixes under the model)**.** *Under the product-uniform model, for any ordering $\pi_c$ and any depth $k$,*

$$\mathbb{E}\big[V_{\pi_c}^{(k)}\big] = g\Big(N, \prod_{i=1}^{k} K_{\pi_i}\Big).$$

*Moreover, for fixed $N$, $g(N, K)$ is strictly increasing in $K$.*

*Proof.* At depth $k$, the length-$k$ prefix is a sample from the uniform distribution over $K_{1:k} := \prod_{i=1}^{k} K_{\pi_i}$ possible keys. Thus $V_{\pi_c}^{(k)}$ equals the number of occupied keys, whose expectation is the standard occupancy formula (24). Monotonicity in $K$ follows since increasing the number of bins (keeping uniform draws) weakly increases the expected number of occupied bins; one can verify by comparing $g(N, K+1) - g(N, K) > 0$. $\square$

**Theorem C.6** (Optimality in expectation under independence)**.** *Under the product-uniform model, an ordering that sorts columns by nondecreasing alphabet size $K_a$ minimizes $\sum_{k=1}^{M} \mathbb{E}[V_{\pi_c}^{(k)}]$, and hence maximizes $\mathbb{E}[\mathcal{H}(\pi_c)]$. Algorithm 1 returns such an ordering (up to ties), and therefore is optimal in expectation:*

$$\mathbb{E}[\mathcal{H}(\pi_c^{\mathrm{SOLO}})] = \max_{\pi_c} \mathbb{E}[\mathcal{H}(\pi_c)].$$

*Proof.* By Lemma C.5,

$$\mathbb{E}[\mathcal{H}(\pi_c)] = \sum_{k=1}^{M}\Big(N - \mathbb{E}[V_{\pi_c}^{(k)}]\Big) = MN - \sum_{k=1}^{M} g\Big(N, \prod_{i=1}^{k} K_{\pi_i}\Big).$$

Since $g(N, \cdot)$ is increasing, maximizing $\mathbb{E}[\mathcal{H}(\pi_c)]$ is equivalent to minimizing the partial products $\prod_{i=1}^{k} K_{\pi_i}$ at every depth $k$. This is achieved by ordering columns by nondecreasing $K_a$: for any adjacent pair with $K_x > K_y$, swapping them decreases the $k$-th partial product (from $L \cdot K_x$ to $L \cdot K_y$ where $L$ is the product of previous sizes) while leaving the $(k+1)$-th partial product unchanged (both equal $L \cdot K_x K_y$), thus strictly improving the objective. Repeatedly applying adjacent swaps yields the globally optimal sorted order.

Finally, at step $k$, Algorithm 1 chooses $a$ minimizing $V^{(k)}$ (equivalently maximizing the savings), and under the model this is equivalent to choosing the smallest remaining $K_a$ because $\mathbb{E}[V^{(k)}] = g(N, L \cdot K_a)$ with fixed $L$ from previous steps and $g$ increasing. Hence greedy returns an optimal ordering (up to ties). $\square$

## D. Supplementary Experimental Results

### D.1. Dataset and Prompt Length Statistics

Table 7 reports dataset sizes and input prompt token-length distributions under our semantic scan template. These statistics contextualize system results, since longer prompts amplify the value of prefix reuse.

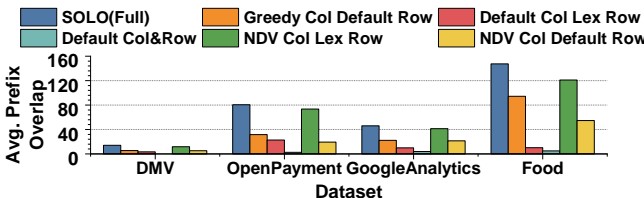

*Figure 10.* Column and row ablation.

## D.2. Ablation Study

**Ablation.** Figure 10 isolates column and row reordering by comparing: Default Col&Row; Greedy Col Default Row (greedy columns, default rows); NDV Col Default Row (NDV columns, default rows); Default Col Lex Row (default columns, lex rows); NDV Col Lex Row (NDV columns, lex rows); and SOLO(Full). Column reordering increases the potential for reuse, but without a compatible row order much of the benefit remains unrealized; only the reuse-aware column order paired with a matching row schedule consistently yields higher prefix overlap.

## D.3. Additional Baselines, Sensitivity, and Fairness Checks

This section gives the full baseline rankings and sensitivity results referenced by the main experiments. All results use the same hardware and serving configuration as the main experiments unless stated otherwise.

**Request granularity.** Semantic operators naturally make independent row-level judgments. Table 8 compares this one-row-per-request design against packing multiple rows into one prompt on OpenPayments. Under the same task setting, one row per request gives the highest accuracy and the lowest per-row latency.

*Table 8.* One-row-per-request vs. multi-row prompt packing on OpenPayments.

| Rows/Prompt | Accuracy | Time/Row |
|---|---|---|
| 1 | **100.0%** | **204ms** |
| 2 | 98.6% | 249ms |
| 4 | 99.1% | 262ms |
| 8 | 99.0% | 307ms |

**Additional baselines.** Table 9 summarizes the simple baseline comparison. SOLO achieves the highest prefix overlap on every dataset, while random ordering degrades below the raw layout and single-key sorting captures only limited prefix locality.

*Table 9.* Additional baselines. Values are prefix overlap; Single-key reports the best tested key.

| Dataset | Default | Random | Single-key | Cluster | FreqLex | NDV | GGR | SOLO |
|---|---|---|---|---|---|---|---|---|
| DMV | 0.80 | 0.54 | 1.83 | 3.00 | 11.16 | 11.16 | 11.65 | **12.72** |
| OpenPayments | 10.49 | 3.36 | 11.32 | 15.94 | 72.48 | 72.47 | 62.72 | **75.93** |
| GoogleAnalytics | 1.16 | 0.37 | 2.16 | 4.06 | 40.68 | 40.68 | 39.51 | **44.68** |
| Food | 0.00 | 0.00 | 1.00 | 2.00 | 140.43 | 140.30 | 99.58 | **154.91** |

**Adjacent overlap under vLLM prefix caching.** The theory optimizes adjacent overlap, while the deployed vLLM cache can reuse matching prefixes that remain in memory across earlier requests. Table 11 shows that adjacent overlap remains a tight surrogate for realized serving outcomes in this cache.

*Table 10.* Full baseline rankings using prefix overlap and prefix-cache hit rate.

*(a)* OpenPayments

| Rank | Method | Overlap | Hit rate |
|---|---|---|---|
| 1 | SOLO | **75.93** | **76.3%** |
| 2 | FreqLex | 72.48 | 66.1% |
| 3 | NDV sort | 72.47 | 66.1% |
| 4 | GGR | 62.72 | 63.9% |
| 5 | Column-only SOLO | 54.47 | 65.3% |
| 6 | ClusterPrefix | 15.94 | 34.2% |
| 7 | Row-only lexicographic | 14.44 | 29.9% |
| 8 | Single-key (min NDV) | 11.32 | 25.8% |
| 9 | Single-key (max freq.) | 11.32 | 25.8% |
| 10 | Single-key (first attr.) | 10.51 | 23.8% |
| 11 | Default layout | 10.49 | 23.7% |
| 12 | Random row order | 3.36 | 7.3% |

*(b)* DMV

| Rank | Method | Overlap | Hit rate |
|---|---|---|---|
| 1 | SOLO | **12.72** | **35.2%** |
| 2 | GGR | 11.65 | 30.9% |
| 3 | FreqLex | 11.16 | 28.8% |
| 4 | NDV sort | 11.16 | 28.8% |
| 5 | Column-only SOLO | 8.26 | 31.4% |
| 6 | ClusterPrefix | 3.00 | 8.6% |
| 7 | Single-key (min NDV) | 1.83 | 6.1% |
| 8 | Single-key (max freq.) | 1.80 | 6.1% |
| 9 | Row-only lexicographic | 1.00 | 1.8% |
| 10 | Single-key (first attr.) | 1.00 | 1.8% |
| 11 | Default layout | 0.80 | 1.8% |
| 12 | Random row order | 0.54 | 0.04% |

*(c)* GoogleAnalytics

| Rank | Method | Overlap | Hit rate |
|---|---|---|---|
| 1 | SOLO | **44.68** | **62.1%** |
| 2 | FreqLex | 40.68 | 56.8% |
| 3 | NDV sort | 40.68 | 56.8% |
| 4 | GGR | 39.51 | 56.9% |
| 5 | Column-only SOLO | 27.02 | 52.5% |
| 6 | ClusterPrefix | 4.06 | 7.4% |
| 7 | Single-key (min NDV) | 2.16 | 6.4% |
| 8 | Single-key (max freq.) | 2.16 | 6.4% |
| 9 | Row-only lexicographic | 2.06 | 4.0% |
| 10 | Single-key (first attr.) | 2.00 | 3.5% |
| 11 | Default layout | 1.16 | 3.2% |
| 12 | Random row order | 0.37 | 1.8% |

*(d)* Food

| Rank | Method | Overlap | Hit rate |
|---|---|---|---|
| 1 | SOLO | **154.91** | **71.1%** |
| 2 | FreqLex | 140.43 | 66.5% |
| 3 | NDV sort | 140.30 | 66.3% |
| 4 | Column-only SOLO | 127.49 | 66.7% |
| 5 | GGR | 99.58 | 47.7% |
| 6 | ClusterPrefix | 2.00 | 0.0% |
| 7 | Single-key (max freq.) | 1.00 | 0.0% |
| 8 | Single-key (min NDV) | 1.00 | 0.0% |
| 9 | Single-key (first attr.) | 0.00 | 0.0% |
| 10 | Row-only lexicographic | 0.00 | 0.0% |
| 11 | Default layout | 0.00 | 0.0% |
| 12 | Random row order | 0.00 | 0.0% |

*Table 11.* Pearson correlation between adjacent overlap and realized serving metrics.

| Dataset | Reused KV | Prefill TPS |
|---|---|---|
| DMV | 0.997 | 0.995 |
| OpenPayments | 0.993 | 0.998 |
| GoogleAnalytics | 1.000 | 0.998 |
| Food | 0.999 | 0.981 |

**Serving sensitivity.** Tables 12–14 separate the sensitivity study into cache budget, batch size, and prompt-prefix length. Gains remain positive at batch size 1 and grow with concurrency.

*Table 12.* KV-cache sensitivity: SOLO hit-rate gain in percentage points.

| KV cache | OpenPayments | Food |
|---|---|---|
| 0.3GB | +69.5 | +60.8 |
| 0.5GB | +69.4 | +61.0 |
| 2.0GB | +69.2 | +61.2 |
| 3.5GB | +68.9 | +61.5 |
| 5.5GB | +68.5 | +61.7 |

*Table 13.* Batch-size sensitivity: default vs. SOLO prefill TPS.

| Batch | OP Default | OP SOLO | Food Default | Food SOLO |
|---|---|---|---|---|
| 1 | 2882 | **5504** | 4133 | **9008** |
| 8 | 5555 | **30188** | 4939 | **13951** |
| 32 | 5937 | **57064** | 5089 | **15202** |
| 128 | 6074 | **91691** | 5102 | **15559** |

*Table 14.* Prompt-prefix sensitivity: SOLO hit-rate gain in percentage points.

| Prompt prefix | OpenPayments | Food |
|---|---|---|
| 5 tokens | +71.7 | +61.9 |
| 50 tokens | +69.4 | +61.0 |
| 200 tokens | +62.5 | +58.1 |
| 500 tokens | +52.3 | +53.2 |

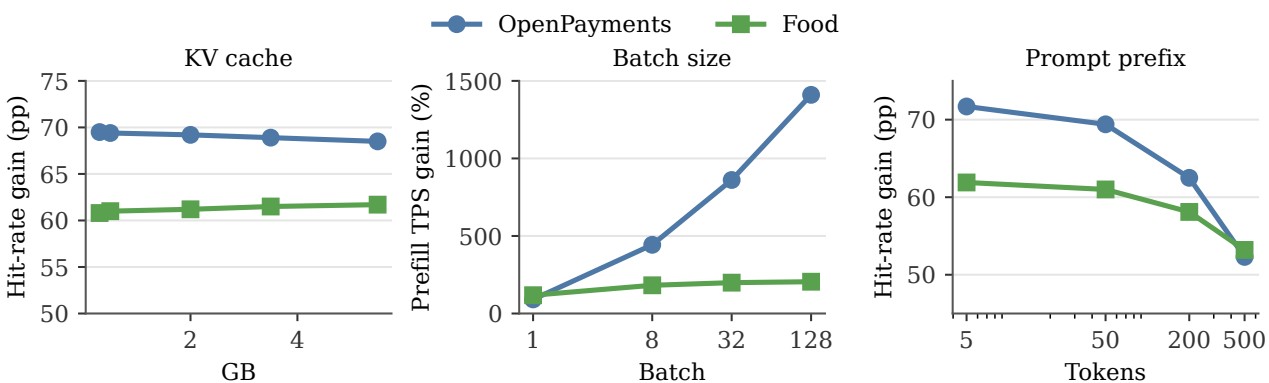

*Figure 11.* Serving sensitivity across KV-cache budget, batch size, and prompt-prefix length.

**Matched planning-time fairness.** Table 15 gives the matched planning-time comparison. SOLO reaches higher overlap in 1.5s than stochastic and recursive baselines reach after a 187.5s ceiling on the same hardware.

*Table 15.* Matched planning-time comparison on OpenPayments. Values are prefix overlap at each elapsed time.

| Method | 1.5s | 10s | 30s | 187.5s |
|--------|------|------|------|--------|
| SOLO | **75.93** | – | – | – |
| ACLR | 6.27 | 26.04 | 37.49 | 54.97 |
| GCLR | 26.11 | 26.11 | 25.78 | 36.85 |
| GGR | – | – | – | 62.72 |

**Materialization cost.** Table 16 separates layout planning from physical row/column reorder. Both costs are small compared with inference savings.

*Table 16.* Materialization and planning overhead at the 100K-tuple scale.

| Dataset | Plan | Reorder | Total | Raw inf. | SOLO inf. | Saved | Plan/Saved | Reorder/Saved |
|---------|------|---------|-------|----------|-----------|-------|------------|---------------|
| DMV | 2.8s | 0.9s | 3.7s | 1.7h | 1.2h | 26min | 0.18% | 0.06% |
| OpenPayments | 14.1s | 3.4s | 17.5s | 7.4h | 1.4h | 6.0h | 0.065% | 0.016% |

**Robustness to correlations and tokenization.** Table 17 evaluates functional-dependency robustness, and Table 18 measures how often shared attribute prefixes become token-level reusable prefixes across tokenizer families. For the injected-FD variants, the FD level denotes the fraction of columns replaced by deterministic transformations of other columns, such as prefix extraction, hash buckets, or fixed lookup mappings. This preserves the table size while creating controlled dependencies.

*Table 17.* Functional-dependency robustness using prefix overlap.

| Data | FD level | Default | NDV | SOLO |
|------|----------|---------|------|------|
| Flight | natural | 3.03 | 57.48 | **73.06** |
| OpenPayments | 0% | 10.49 | 72.47 | **75.93** |
| OpenPayments | 20% | 14.10 | 52.47 | **73.75** |
| OpenPayments | 50% | 14.26 | 47.38 | **73.03** |
| OpenPayments | 80% | 26.58 | 45.26 | **65.55** |

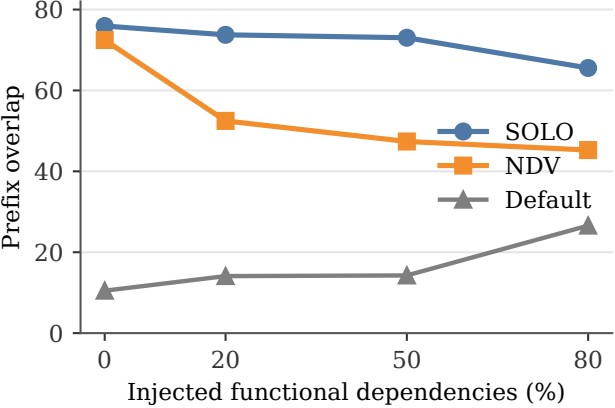

*Figure 12.* Functional-dependency stress test on OpenPayments using prefix overlap.

*Table 18.* Tokenizer robustness: token-level reuse ratio from shared attribute prefixes.

| Dataset | #Cols | Qwen2.5-BPE | GPT-2-BPE | Mistral-SP |
|---|---|---|---|---|
| DMV | 20 | 84.5% | 84.3% | 89.1% |
| OpenPayments | 90 | 91.3% | 97.1% | 95.5% |
| GoogleAnalytics | 54 | 87.3% | 90.3% | 90.5% |
| Food | 200 | 86.2% | 97.6% | 93.3% |
| Flight | 110 | 90.7% | 91.1% | 91.2% |

**Planner scalability.** Tables 19 and 20 report peak memory and end-to-end planning time on Food-derived stress tests.

*Table 19.* Peak planner memory.

| $M = 200$ | | $N = 10^7$ | |
|---|---|---|---|
| $N$ | Memory | $M$ | Memory |
| $10^5$ | 0.32GB | 50 | 2.10GB |
| $10^6$ | 0.88GB | 100 | 3.96GB |
| $5 \times 10^6$ | 3.88GB | 150 | 5.83GB |
| $10^7$ | 7.67GB | 200 | 7.69GB |

*Table 20.* End-to-end planning time for column ordering plus lexicographic row sorting.

| $M = 200$ | | $N = 10^7$ | |
|---|---|---|---|
| $N$ | Time | $M$ | Time |
| $10^5$ | 30s | 20 | 24s |
| $10^6$ | 36s | 50 | 51s |
| $5 \times 10^6$ | 86s | 100 | 105s |
| $10^7$ | 180s | 150 | 143s |
| | | 200 | 183s |

# E. Small-scale Comparison to OPHR

**Small-scale exact-search comparison.** OPHR (Liu et al., 2025b) can be interpreted as searching over both a row permutation and a *per-tuple* column permutation. In the worst case, this corresponds to enumerating $N!$ row orders and $(M!)^N$ per-tuple column orders, i.e.,

$$\mathcal{O}\big(N!\,(M!)^N\big),\tag{25}$$

which becomes intractable even for moderate $N$ and $M$. In contrast, SOLO outputs a *global* column order via a greedy procedure and then sorts rows lexicographically under that column order, with time complexity

$$\mathcal{O}\big(M^2 N + M N \log N\big).\tag{26}$$

This exponential search makes OPHR useful as an exact-search reference on small instances. We compare OPHR and SOLO on instances that finish under a fixed wall-clock budget.

**Protocol (small-$N$/small-$M$ on a real table).** We run OPHR and SOLO on small sub-tables extracted from a real structured dataset (DMV). For each instance with $N$ tuples and $M$ attributes, both methods output a reordering plan, and we measure:

- **Field-level prefix overlap** (higher is better): we linearize each tuple using the same fixed template as the main experiments and define $\text{LCP}_{\text{field}}(i-1, i)$ as the number of *consecutive attribute–value fields* (under the chosen global column order) that match between two adjacent tuples $i-1$ and $i$ (excluding the constant instruction prefix). We report

$$\text{Overlap} \;=\; \frac{1}{N-1}\sum_{i=2}^{N} \text{LCP}_{\text{field}}(i-1, i),\tag{27}$$

| Scenario | $N$ | $M$ | Overlap↑ | | Time (s)↓ | |
|---|---|---|---|---|---|---|
| | | | SOLO (ours) | OPHR | SOLO (ours) | OPHR |
| M-Scale | 5 | 4 | 1.0000 | **1.0000** | 0.6763 | 1.2042 |
| M-Scale | 5 | 5 | 1.5000 | **1.7500** | 0.6426 | 2.3722 |
| M-Scale | 5 | 8 | 1.5000 | **1.7500** | 0.6375 | 4.8063 |
| N-Scale | 10 | 3 | 1.1889 | **1.5556** | 0.6329 | 40.4162 |
| N-Scale | 12 | 3 | 1.2727 | **1.5455** | 0.7352 | 169.2988 |
| N-Scale | 14 | 3 | 1.3846 | **1.6154** | 0.7301 | 755.0047 |
| Deep-N | 12 | 2 | 0.7273 | **0.8182** | 0.7070 | 104.4999 |
| Deep-N | 15 | 2 | 0.7143 | **0.8571** | 0.7200 | 997.3976 |
| Hybrid | 12 | 4 | 1.2727 | **1.5455** | 0.6245 | 301.3156 |
| Hybrid | 13 | 4 | 1.3333 | **1.5833** | 0.7254 | 644.1272 |

*Table 21.* Small-scale OPHR comparison on real sub-tables from `DMV.csv`. OPHR gives the best overlap on finished runs; SOLO remains sub-second.

| Scenario | $N$ | $M$ | OPHR time (s) |
|---|---|---|---|
| Hybrid | 14 | 4 | **TIMEOUT** ($>$1200) |
| Hybrid | 20 | 6 | **TIMEOUT** ($>$1200) |
| Hybrid | 25 | 8 | **TIMEOUT** ($>$1200) |
| Hybrid | 30 | 10 | **TIMEOUT** ($>$1200) |

*Table 22.* OPHR planning time under the 1200s cap. TIMEOUT marks unfinished runs.

which lies in $[0, M]$ and isolates the effect of layout optimization from tokenizer-dependent token counts.

- **Planning Overhead** (lower is better): wall-clock time (CPU) to compute the reordering plan, excluding LLM execution.

**Results.** Table 21 reports representative settings. When OPHR finishes, it attains the best overlap, while its planning time grows quickly: OPHR is $\sim 10^3 \times$ slower than SOLO at ($N$=14, $M$=3) and $\sim 1.4 \times 10^3 \times$ slower at ($N$=15, $M$=2). At larger settings (Table 22), OPHR reaches the 1200s cap, while SOLO remains sub-second.

## F. Experimental Setup and Reproducibility

**Hardware and Serving Environment.** Experiments are conducted on a server equipped with $4\times$ NVIDIA RTX A6000 GPUs, 40 Intel Xeon Silver 4210R CPU cores, and 504GB RAM. We utilize `vLLM` v0.10.0 with prefix caching enabled, enforcing a uniform configuration across all methods to ensure strict fairness: `gpu_memory_utilization=0.66`, `max_model_len=4096`, block size 16, and an LRU eviction policy. This configuration yields effective KV cache budgets of 74,320 tokens for Qwen2.5-7B and 12,528 tokens for Qwen2.5-14B. All approaches share identical batching policies, tokenizer settings, and scheduler limits; thus, reported performance differences are strictly attributable to the ordering strategy rather than system-level variances.

**Models and Baselines.** Semantic operators utilize Qwen2.5-14B (Qwen et al., 2024), while proxy-based filtering employs Qwen2.5-0.5B (proxy) and Qwen2.5-14B (oracle). Baseline hyperparameters are fixed for reproducibility: Genetic Algorithm (GA) uses `pop_size=30`, `gens=10`, `mut_rate=0.2`, `tourn_k=3`, and `rng_seed=42` (totaling 300 evaluations); Simulated Annealing (SA) sets `iters=200`, `T0=1.0`, `alpha=0.995`, and `rng_seed=123`. Deterministic methods (Vortex, Refgc, Z-gray) are parameter-free. Finally, when evaluating column-only ordering methods, we apply a fixed row ordering to isolate gains solely attributable to column permutation.

## G. Worst-Case Runtime of Greedy Group Recursion (GGR)

This appendix derives an explicit worst-case runtime upper bound for the Greedy Group Recursion (GGR) heuristic in Liu et al. (2025b).

## G.1. Preliminaries

Let the input table have $N$ rows and $M$ columns. During recursion, GGR is invoked on subtables of size $n \times m$, where $1 \leq n \leq N$ and $1 \leq m \leq M$. GGR evaluates, at each recursion step, all candidates $(c, v)$ where $c$ ranges over columns and $v$ ranges over $\mathrm{distinct}(T[:, c])$ (lines 17–23), calls HITCOUNT for each candidate (lines 3–8), selects the best, and then recurses on two subtables (lines 24–26).

We analyze the asymptotic cost implied by these algorithmic steps. In particular, enumerating $\mathrm{distinct}(T[:, c])$ and constructing $R_v = \{i \mid T[i, c] = v\}$ are scanning-based operations over the current subtable, consistent with the paper's note that per-step scanning for distinct values can yield quadratic cost in terms of table size.

## G.2. Cost of HITCOUNT

**Lemma G.1** (Cost of HITCOUNT). *On an $n \times m$ subtable,* HITCOUNT *runs in $\mathcal{O}(mn)$ time in the worst case.*

*Proof.* GGR defines $R_v \leftarrow \{i \mid T[i, c] = v\}$ (GGR's line 4), which requires scanning the $n$ rows of column $c$ and thus costs $\mathcal{O}(n)$. It then forms $inferred\_cols \leftarrow \{c' \mid (c, c') \in FD\}$ (lines 5–6), whose size is at most $m - 1$. Finally, it computes $\sum_{c' \in inferred\_cols} \sum_{r \in R_v} \mathrm{len}(T[r, c'])$ (GGR's line 6), which is $\mathcal{O}(|inferred\_cols| \cdot |R_v|) \subseteq \mathcal{O}(mn)$. All remaining arithmetic is constant-time. Therefore, HITCOUNT is $\mathcal{O}(mn)$ in the worst case. $\square$

## G.3. Non-recursive Cost per GGR Call

**Lemma G.2** (Per-call selection cost). *For $n, m \geq 2$, the non-recursive work of a single* GGR *call on an $n \times m$ subtable is $\mathcal{O}(m^2 n^2)$ in the worst case.*

*Proof.* At the selection step (GGR's lines 17–23), GGR iterates over all columns $c$ and all values $v \in \mathrm{distinct}(T[:, c])$. In the worst case, each column has $n$ distinct values, so the number of candidate pairs is at most $mn$. For each candidate, GGR invokes HITCOUNT, which costs $\mathcal{O}(mn)$ by Lemma G.1. Hence the selection step costs $\mathcal{O}(mn) \cdot \mathcal{O}(mn) = \mathcal{O}(m^2 n^2)$ in the worst case. $\square$

## G.4. Recurrence

Let $T(n, m)$ denote the worst-case runtime of GGR on an $n \times m$ subtable. For base cases, if $n = 1$ or $m = 1$, the algorithm terminates after at most a scan of the remaining entries, so $T(1, m) = \mathcal{O}(m)$ and $T(n, 1) = \mathcal{O}(n)$ suffice for our bound.

For $n, m \geq 2$, after selecting $(b_c, b_v)$, the algorithm forms $R_v$ (GGR's line 24) and recurses on: (i) $T[\mathrm{rows} \setminus R_v,\ \mathrm{cols}]$ of size $(n - k) \times m$ where $k := |R_v| \in \{1, \ldots, n - 1\}$, and (ii) $T[R_v,\ \mathrm{cols} \setminus b\_cols]$ of size $k \times m'$ where $m' \leq m - 1$ because $b\_cols$ contains at least the pivot column $b_c$ (GGR's line 26). Thus, for some $k \in \{1, \ldots, n - 1\}$,

$$T(n, m) \ \leq \ T(n - k, m) \ + \ T(k, m - 1) \ + \ C\, m^2 n^2, \tag{28}$$

where the last term upper bounds the per-call non-recursive work by Lemma G.2 and $C > 0$ is a constant.

## G.5. Main Result

**Theorem G.3** (Worst-case runtime of GGR). *The worst-case runtime of* GGR *on an $N \times M$ table is*

$$T(N, M) \ = \ \mathcal{O}\big(M^2 N^2 (M + N)\big).$$

*Proof.* We prove by induction on $n + m$ that there exist constants $A, B > 0$ such that for all $n, m \geq 1$,

$$T(n, m) \ \leq \ A\, m^2 n^3 \ + \ B\, m^3 n^2. \tag{29}$$

Since $m^2 n^3 + m^3 n^2 = m^2 n^2 (m + n)$, the theorem follows by substituting $(n, m) = (N, M)$.

**Base cases.** If $n = 1$, then $T(1, m) = \mathcal{O}(m) \leq Am^2 + Bm^3$ for sufficiently large $A, B$. If $m = 1$, then $T(n, 1) = \mathcal{O}(n) \leq An^3 + Bn^2$ for sufficiently large $A, B$. Thus (29) holds for $n + m \leq 2$.

| Method | Prefill TPS ($\uparrow$) | Speedup | TTFT (s) ($\downarrow$) | TTFT$\downarrow$ | Reuse ratio ($\uparrow$) | F1 ($\uparrow$) | Reorder (ms/req) |
|---|---|---|---|---|---|---|---|
| NATURAL | $8887.99 \pm 44.63$ | $1.000\times$ | $0.0248 \pm 0.0001$ | 0.0% | $0.4926 \pm 0.0000$ | $0.2538 \pm 0.0573$ | $0.0001 \pm 0.0000$ |
| RANDOM | $8799.71 \pm 54.24$ | $0.990\times$ | $0.0251 \pm 0.0002$ | -1.0% | $0.4912 \pm 0.0000$ | $0.2253 \pm 0.0280$ | $0.0006 \pm 0.0002$ |
| SOLO | $10656.31 \pm 85.41$ | $1.199\times$ | $0.0207 \pm 0.0002$ | 16.6% | $0.5886 \pm 0.0000$ | $0.2100 \pm 0.0224$ | $0.0584 \pm 0.0230$ |

*Table 23.* Walmart–Amazon entity matching with vLLM prefix caching.

**Inductive step.** Assume (29) holds for all $(n', m')$ with $n' + m' < n + m$, and consider $(n, m)$ with $n, m \geq 2$. Let $k \in \{1, \ldots, n-1\}$ be the size of $R_v$ at this call. Applying the recurrence (28) and the induction hypothesis to the two subproblems gives:

$$
\begin{aligned}
T(n, m) &\leq T(n-k, m) + T(k, m-1) + Cm^2 n^2 \\
&\leq \left( Am^2(n-k)^3 + Bm^3(n-k)^2 \right) + \left( A(m-1)^2 k^3 + B(m-1)^3 k^2 \right) + Cm^2 n^2.
\end{aligned}
\tag{30}
$$

Using $(m-1)^2 \leq m^2$ and $(m-1)^3 \leq m^3$, we obtain

$$
T(n, m) \leq Am^2\big((n-k)^3 + k^3\big) + Bm^3\big((n-k)^2 + k^2\big) + Cm^2 n^2.
\tag{31}
$$

For nonnegative $x, y$, we have $x^3 + y^3 \leq (x+y)^3$ and $x^2 + y^2 \leq (x+y)^2$. Letting $x = n - k$ and $y = k$ yields

$$
(n-k)^3 + k^3 \leq n^3, \qquad (n-k)^2 + k^2 \leq n^2.
$$

Substituting into (31) gives

$$
T(n, m) \ \leq \ Am^2 n^3 \ + \ Bm^3 n^2 \ + \ Cm^2 n^2.
\tag{32}
$$

Since $n \geq 1$ implies $n^3 \geq n^2$, we have $Cm^2 n^2 \leq Cm^2 n^3$. Choosing $A$ large enough so that $A \geq C$ allows absorbing the last term into the first, preserving the form of (29). This completes the induction. $\square$

## H. Table-Structured Prompt Optimization: Entity Matching

**Experimental Setup**   We evaluate SOLO on Entity Matching (EM), a canonical AI task that determines if two structured records (e.g., product specs) refer to the same entity (Getoor & Machanavajjhala, 2012). Using the Walmart–Amazon benchmark ($N = 2000$ pairs) (Satrio, 2019), we prompt Qwen2.5-7B-Instruct (served via vLLM with prefix caching) to classify record pairs as MATCH or MISMATCH. We compare three field serialization strategies: (1) NATURAL (original dataset order); (2) RANDOM (random permutation); and (3) SOLO (reordering commutative fields to maximize global prefix reuse).

**Results**   Table 23 presents the systems and quality metrics. SOLO increases the KV-cache reuse ratio from 0.49 to 0.59, giving a $1.20\times$ speedup in prefill throughput and a 16.6% reduction in Time-To-First-Token (TTFT), with 0.06 ms/req reordering overhead. The table reports F1 alongside serving metrics so the efficiency gains can be read with task quality.

## I. Quality Preservation Under Layout Reordering

**Experimental Setup**   We evaluate whether layout reordering changes record-level task outcomes on a fuzzy value matching task. We compare the original layout against three reordering strategies: (1) Row-only (lexicographical, random); (2) Column-only (NDV, SOLO, random); and (3) Combined (NDV columns plus lexicographical rows, SOLO columns plus lexicographical rows, GGR layout, random full layout). We keep content, prompt template, and model fixed, and report mean and standard deviation over five independent repetitions.

**Metrics: Paired Outcomes and Stability**   For each record, we compare the original-layout prediction, the reordered-layout prediction, and the ground-truth label. The paired outcomes are: both layouts correct, both layouts wrong, corrected by reordering, and regressed after reordering. From these outcomes, we report changed correctness, net accuracy change, and unchanged correctness.

**Results** Tables 24–27 summarize the results. Row reordering changes correctness by at most $0.5\%$. For SOLO column-only and combined layouts, the worst absolute accuracy change is $0.6\%$, changed correctness is at most $3.0\%$, and unchanged correctness is at least $97.0\%$. These results place the serving gains in a fixed-content task setting.

*Table 24.* Quality preservation on DMV (percentages; mean±std over five runs).

| Group | Variant | Both correct | Both wrong | Corrected | Regressed | Changed correctness | Accuracy change | Unchanged correctness |
|---|---|---|---|---|---|---|---|---|
| Column | NDV order | 97.8±1.6 | 1.0±0.8 | 1.0±0.6 | 1.6±1.1 | 2.6±1.2 | -0.6±1.3 | 97.4±1.2 |
| Column | SOLO order | 98.5±1.1 | 0.8±0.6 | 1.3±0.9 | 0.9±0.7 | 2.2±1.0 | 0.4±1.1 | 97.8±1.0 |
| Column | Random order | 97.9±1.3 | 1.1±0.7 | 1.3±0.6 | 1.1±0.9 | 2.4±1.0 | 0.2±1.0 | 97.6±1.0 |
| Combined | NDV columns + lex rows | 97.5±1.6 | 1.1±0.8 | 1.1±0.7 | 2.3±1.2 | 3.4±1.3 | -1.2±1.4 | 96.6±1.3 |
| Combined | SOLO columns + lex rows | 98.3±1.2 | 0.8±0.6 | 1.4±0.9 | 1.0±0.7 | 2.4±1.0 | 0.4±1.1 | 97.6±1.0 |
| Combined | GGR layout | 98.5±1.0 | 0.4±0.4 | 1.5±0.7 | 0.6±0.5 | 2.1±0.9 | 0.9±0.9 | 97.9±0.9 |
| Combined | Random full layout | 97.9±1.2 | 1.2±0.8 | 1.1±0.6 | 1.8±1.0 | 2.9±1.1 | -0.7±1.1 | 97.1±1.1 |
| Row | Lexicographic rows | 98.8±0.8 | 1.2±0.8 | 0.0±0.0 | 0.0±0.0 | 0.0±0.0 | 0.0±0.0 | 100.0±0.0 |
| Row | Random rows | 98.8±0.8 | 1.2±0.8 | 0.0±0.0 | 0.0±0.0 | 0.0±0.0 | 0.0±0.0 | 100.0±0.0 |

*Table 25.* Quality preservation on OpenPayments (percentages; mean±std over five runs).

| Group | Variant | Both correct | Both wrong | Corrected | Regressed | Changed correctness | Accuracy change | Unchanged correctness |
|---|---|---|---|---|---|---|---|---|
| Column | NDV order | 99.1±0.5 | 0.5±0.3 | 0.3±0.3 | 0.2±0.2 | 0.5±0.4 | 0.2±0.4 | 99.5±0.4 |
| Column | SOLO order | 98.4±1.1 | 0.4±0.3 | 0.4±0.5 | 0.8±1.0 | 1.2±1.3 | -0.4±0.9 | 98.8±1.3 |
| Column | Random order | 98.5±0.4 | 0.6±0.4 | 0.2±0.2 | 0.7±0.3 | 0.9±0.4 | -0.5±0.4 | 99.1±0.4 |
| Combined | NDV columns + lex rows | 98.7±0.6 | 0.4±0.3 | 0.5±0.4 | 0.4±0.3 | 0.9±0.6 | 0.2±0.5 | 99.1±0.6 |
| Combined | SOLO columns + lex rows | 98.6±0.8 | 0.3±0.2 | 0.6±0.6 | 0.8±0.9 | 1.4±0.9 | -0.2±0.9 | 98.6±0.9 |
| Combined | GGR layout | 98.2±0.6 | 0.3±0.2 | 0.6±0.6 | 0.9±0.3 | 1.5±0.8 | -0.2±0.6 | 98.5±0.8 |
| Combined | Random full layout | 98.5±0.6 | 0.5±0.4 | 0.5±0.3 | 0.5±0.6 | 1.0±0.7 | -0.1±0.6 | 99.0±0.7 |
| Row | Lexicographic rows | 99.5±0.5 | 0.0±0.0 | 0.3±0.5 | 0.2±0.2 | 0.5±0.5 | 0.1±0.4 | 99.5±0.5 |
| Row | Random rows | 99.7±0.4 | 0.0±0.0 | 0.1±0.2 | 0.2±0.3 | 0.3±0.4 | -0.2±0.3 | 99.7±0.4 |

*Table 26.* Quality preservation on GoogleAnalytics (percentages; mean±std over five runs).

| Group | Variant | Both correct | Both wrong | Corrected | Regressed | Changed correctness | Accuracy change | Unchanged correctness |
|---|---|---|---|---|---|---|---|---|
| Column | NDV order | 99.3±0.4 | 0.2±0.2 | 0.0±0.0 | 0.5±0.2 | 0.5±0.2 | -0.5±0.2 | 99.5±0.2 |
| Column | SOLO order | 99.4±0.4 | 0.0±0.0 | 0.0±0.0 | 0.6±0.2 | 0.6±0.2 | -0.6±0.2 | 99.4±0.2 |
| Column | Random order | 99.5±0.3 | 0.2±0.2 | 0.0±0.0 | 0.3±0.2 | 0.3±0.2 | -0.3±0.2 | 99.7±0.2 |
| Combined | NDV columns + lex rows | 99.6±0.3 | 0.0±0.0 | 0.0±0.0 | 0.4±0.3 | 0.4±0.3 | -0.4±0.3 | 99.6±0.3 |
| Combined | SOLO columns + lex rows | 99.4±0.4 | 0.0±0.0 | 0.0±0.0 | 0.6±0.2 | 0.6±0.2 | -0.6±0.2 | 99.4±0.2 |
| Combined | GGR layout | 99.6±0.4 | 0.0±0.0 | 0.1±0.2 | 0.4±0.3 | 0.5±0.3 | -0.4±0.3 | 99.5±0.3 |
| Combined | Random full layout | 99.8±0.2 | 0.0±0.0 | 0.0±0.0 | 0.2±0.2 | 0.2±0.2 | -0.2±0.2 | 99.8±0.2 |
| Row | Lexicographic rows | 99.8±0.2 | 0.0±0.0 | 0.2±0.2 | 0.0±0.0 | 0.2±0.2 | 0.2±0.2 | 99.8±0.2 |
| Row | Random rows | 99.8±0.2 | 0.0±0.0 | 0.1±0.1 | 0.1±0.2 | 0.2±0.2 | 0.1±0.1 | 99.8±0.2 |

| Dataset | Prompt Len | Out Len | Prefill Mean (s) | Decode Mean (s) | Total Mean (s) | Prefill Share |
|---|---|---|---|---|---|---|
| DMV | 225 | 2 | 0.04 | 0.03 | 0.08 | 59.42% |
| Food | 3120 | 2 | 0.57 | 0.03 | 0.60 | 94.80% |
| GoogleAnalytics | 591 | 2 | 0.11 | 0.03 | 0.14 | 78.36% |
| OpenPayments | 1478 | 2 | 0.26 | 0.03 | 0.29 | 89.52% |

*Table 28.* Per-request timing of a semantic filter operator in LOTUS (mean values).

*Table 27.* Quality preservation on Food (percentages; mean±std over five runs).

| Group | Variant | Both correct | Both wrong | Corrected | Regressed | Changed correctness | Accuracy change | Unchanged correctness |
|---|---|---|---|---|---|---|---|---|
| Column | NDV order | 93.6±1.4 | 3.1±1.0 | 1.8±0.7 | 1.5±1.2 | 3.3±0.7 | 0.3±0.7 | 96.7±0.7 |
| Column | SOLO order | 93.5±1.6 | 3.5±1.4 | 1.4±0.8 | 1.6±1.3 | 3.0±1.1 | -0.2±1.2 | 97.0±1.1 |
| Column | Random order | 94.8±1.6 | 3.3±1.2 | 1.6±0.6 | 0.3±0.2 | 2.0±0.6 | 1.3±0.6 | 98.0±0.6 |
| Combined | NDV columns + lex rows | 93.8±1.5 | 3.4±1.4 | 1.5±1.1 | 1.2±1.1 | 2.7±0.7 | 0.2±1.2 | 97.3±0.7 |
| Combined | SOLO columns + lex rows | 93.8±1.5 | 3.7±1.4 | 1.2±0.9 | 1.3±1.3 | 2.6±1.1 | -0.1±1.2 | 97.4±1.1 |
| Combined | GGR layout | 94.5±1.3 | 3.6±0.8 | 1.3±0.7 | 0.5±0.3 | 1.9±0.6 | 0.8±0.8 | 98.1±0.6 |
| Combined | Random full layout | 94.6±1.7 | 3.2±1.4 | 1.7±0.9 | 0.5±0.5 | 2.2±0.7 | 1.2±0.9 | 97.8±0.7 |
| Row | Lexicographic rows | 94.8±0.3 | 4.7±0.3 | 0.3±0.3 | 0.2±0.2 | 0.5±0.4 | 0.2±0.3 | 99.5±0.4 |
| Row | Random rows | 95.0±0.0 | 4.6±0.0 | 0.4±0.0 | 0.0±0.0 | 0.4±0.0 | 0.4±0.0 | 99.6±0.0 |

# J. Prefill-Dominated Semantic Filter

Semantic operators (e.g., semantic filtering) are *prefill-dominated*: their latency is primarily determined by processing the input prompt (i.e., constructing the KV values and producing the first token), rather than by autoregressive decoding. This behavior follows directly from how semantic operators are invoked. Each invocation must ingest a row-serialized prompt that includes operator instructions together with schema attributes and values, whereas the model output is a compact decision label. Consequently, prefill grows with prompt length, while decoding contributes a smaller and comparatively stable per-request overhead.

**Measurements.** Table 28 reports per-request latency in LOTUS, decomposed into **Prefill** (`scheduled→first_token`), **Decode** (`first_token→last_token`), and **Total** (`scheduled→last_token`) across four datasets: **DMV**, **Open-Payments**, **Food**, and **GoogleAnalytics**. Prompt length and output length are shown for context. We compute $\text{PREFILLSHARE} \triangleq T_{\text{prefill}}/(T_{\text{prefill}} + T_{\text{decode}})$ to quantify how much of the model-side time is attributable to prefill.

Across all datasets, the average output length is consistently 2 tokens (a decision token `T/F` followed by `EOS`), rendering decode time negligible and nearly invariant. Table 28 confirms this: despite significant prompt variation (225–3120 tokens), mean decode time remains constant at $\sim 0.03$ seconds. Prefill varies widely and dominates end-to-end latency. In contrast, prefill time scales with prompt length, ranging from 0.04 seconds (DMV) to 0.57 seconds (Food)—a 12.6× spread. Consequently, PREFILLSHARE accounts for the majority of model-side latency (59.42%–94.80%). This identifies prefill as the governing scaling factor for throughput and latency in non-trivial prompts.

**Implication.** As semantic operators are prefill-bound, system optimizations must prioritize amortizing prompt ingestion and KV construction costs (e.g., through prefix reuse, cache locality, and shared execution strategies).

