# OpenReview forum: "Prefix-Cache-Aware Data Reordering for LLM-Augmented Database Analytics"
_ICML.cc/2026/Conference — ICML 2026 regular_

### Official Review · Reviewer_UyFA · 2026-02-18

**Soundness:** 3
**Presentation:** 3
**Significance:** 4
**Originality:** 4
**Overall Recommendation:** 4
**Confidence:** 4

**Summary:**

The paper proposes SOLO (Semantic Operator Layout Optimizer), a framework to accelerate LLM-based database analytics by maximizing prefix KV-cache reuse. The authors identify that the standard row-by-row processing of relational tables leads to fragmented prompts, preventing effective cache utilization. They formulate the problem of prompt layout optimization as finding paths on a Radix Tree. The paper proves that while optimal column ordering is NP-hard, row ordering can be solved via lexicographical sorting. A greedy tree-shaping algorithm is introduced with a tight approximation guarantee. Experimental results on datasets like OpenPayments and DMV show up to 90.3% throughput improvement and massive reductions in planning overhead compared to prior art like GGR and OPHR.

**Compliance With Llm Reviewing Policy:**

Affirmed.

**Final Justification:**

My concerns have been adequately addressed.

**Key Questions For Authors:**

1. How does the greedy algorithm perform on datasets that exhibit extreme functional dependencies? In such scenarios, does the heuristic sometimes fail to recover the optimal tree structure?
2. Can you share a latency breakdown that explicitly accounts for the time spent physically reordering rows and columns in memory prior to prompt construction? Is this cost negligible for the 100K row scale?
3. Does SOLO need explicit separator tokens to guarantee that Value_A and Value_B are tokenized in a consistent way across different rows?
4. What is the smallest batch size at which SOLO yields a net positive benefit?

**Limitations:**

The authors focus on throughput in batch processing settings. They do not extensively discuss the limitations regarding real-time, low-latency stream processing where batching is not feasible. Additionally, the independence assumption in the theoretical section is a limitation that bounds the theoretical guarantees.

**Strengths And Weaknesses:**

Strengths:
S1: The authors provide rigorous proofs for the NP-hardness of the column ordering problem and the optimality of lexicographical row sorting under a fixed column order.
S2: The reported performance gains are impressive. Achieving a 90.3% improvement in prefill throughput addresses a critical bottleneck in LLM-DB systems.
S3: The proposed solution is designed as a plug-and-play planner that integrates seamlessly into existing systems, such as LOTUS, without requiring modification of the underlying LLM.

Weaknesses:
W1: The theoretical optimality of the greedy algorithm relies on the assumption that columns are mutually independent. In practice, however, relational datasets often exhibit strong functional dependencies and correlations. The paper does not sufficiently analyze how strictly the greedy algorithm's performance degrades when these correlations are strong and the independence assumption is violated.
W2: While the paper extensively discusses planning overhead, it overlooks the data materialization cost. Reordering 100K+ rows and multiple columns requires memory shuffling. In high-throughput, low-latency DB scenarios, the cost of physically moving data to match the new layout might offset the prefill savings.
W3: The effectiveness of prefix caching is highly sensitive to tokenization boundaries. If the tokenizer merges characters across the boundary of two attributes (e.g., the last digit of a number and the first letter of the next field), the prefix might change even if the values are identical, breaking the cache hit. The paper does not discuss if SOLO requires specific tokenizer behaviors to guarantee cache hits.
W4: The approach targets scan-heavy workloads where large batches of tuples are processed. However, in transactional OLTP or streaming scenarios with small batch sizes, the opportunity for reordering is limited. The paper lacks a sensitivity analysis on batch size. At what batch size does the overhead of SOLO exceed its benefits?
W5: The paper compares against heuristic methods like GGR and OPHR. It would be beneficial to compare against more traditional database compression ordering techniques (beyond Vortex) or simple heuristics like sort by cardinality.

---

> ### Author Rebuttal · Authors · 2026-03-31
>
> # Response to Reviewer UyFA
>
> **W1  (Robustness under functional dependencies).**
>
> **SOLO degrades gracefully under extreme functional dependencies** — overlap declines only 14% at 80% FD injection versus 38% for NDV, while the greedy algorithm may not always recover the global optimum, the 1/(M−1) guarantee (Theorem 3.4) ensures it does not degrade significantly from the optimum for any distribution.
>
> We validated this on a real FD benchmark — flight, a 110-column, 100K-row airline dataset (Bureau of Transportation Statistics) widely used in FD discovery research (Papenbrock & Naumann, SIGMOD '16), containing pervasive natural FDs such as Origin→OriginCity→OriginState — and on synthetic FD injection on OpenPayments (90 columns). For injection, we replace 20–80% of columns with deterministic derivations of other columns (e.g., prefix, hash, lookup table), creating controlled functional dependencies while preserving data scale:
>
> | Data | FD Level | Default | NDV | **SOLO** |
> |------|:---:|:---:|:---:|:---:|
> | flight | natural (982K FDs) | 3.03 | 57.48 | **73.06** |
> | OpenPayments | 0% (normal) | 10.49 | 72.47 | **75.93** |
> | OpenPayments | 20% injected | 14.10 | 52.47 | **73.75** |
> | OpenPayments | 50% injected | 14.26 | 47.38 | **73.03** |
> | OpenPayments | 80% injected | 26.58 | 45.26 | **65.55** |
>
> All values are prefix overlap (higher = better).
>
> SOLO achieves the highest overlap on every row. On flight (110 columns with pervasive natural FDs), SOLO reaches 73.06 vs NDV's 57.48. On OpenPayments (90 columns), the contrast is even more striking: under FD injection, NDV's overlap drops sharply (72.47 → 45.26, a 38% decline at 80% FD) because its cardinality-based ordering is disrupted by the correlated columns. SOLO degrades much more gracefully (75.93 → 65.55, a 14% decline), and its advantage over NDV actually *increases* under FD — from +3.5% at 0% to +25.7% at 50%. SOLO's greedy tree-shaping adapts to the data distribution at each depth, whereas static cardinality sorting cannot account for inter-column dependencies.
>
> **W2 (Data materialization cost at 100K). Yes, the materialization cost is negligible at the 100K scale.** We provide the requested latency breakdown below, separating layout planning and physical reorder:
>
> | Dataset | Layout Planning | Phys. Reorder | Total Overhead | Default Inference | SOLO Inference | Time Savings | Planning / Savings | Reorder / Savings |
> |---------|:---:|:---:|:---:|:---:|:---:|:---:|:---:|:---:|
> | DMV 100K | 2.8s | 0.9s | 3.7s | 1.7h | 1.2h | 26 min | 0.18% | 0.06% |
> | OpenPayments 100K | 14.1s | 3.4s | 17.5s | 7.4h | 1.4h | 6.0h | 0.065% | 0.016% |
>
>
> Both planning and materialization costs are negligible relative to the savings (< 0.2%), confirming that data reordering overhead is not a bottleneck at the 100K scale.
>
> **W3 (Tokenizer sensitivity). Yes, separator tokens are needed, but no special tokenizer configuration is required.** To convey tabular semantics to the LLM, standard prompt templates and Markdown table serialization already use delimiters such as `|` to separate attributes. Mainstream tokenizers (BPE, SentencePiece) generally treat `|` as a standalone token, naturally creating clean tokenization boundaries.
>
> We verified this across three tokenizer families (Qwen2.5-BPE, GPT-2-BPE, Mistral-SentencePiece) and five datasets. We report **token-level reuse ratio**: (actual reused KV tokens) / (tokens spanned by the shared attribute-level prefix), i.e., the fraction of the attribute-level overlap that translates to real cache hits:
>
> | Dataset | #Cols | Qwen2.5 (BPE) | GPT-2 (BPE) | Mistral (SentencePiece) |
> |---------|:---:|:---:|:---:|:---:|
> | DMV | 20 | 84.5% | 84.3% | 89.1% |
> | OpenPayments | 90 | 91.3% | 97.1% | 95.5% |
> | GoogleAnalytics | 54 | 87.3% | 90.3% | 90.5% |
> | Food | 200 | 86.2% | 97.6% | 93.3% |
> | flight | 110 | 90.7% | 91.1% | 91.2% |
>
> Across all 15 combinations, 84–98% of attribute-level overlap translates to actual token-level cache reuse. Cross-tokenizer variance within each dataset is <5%.
>
> **W4 (Batch size sensitivity). SOLO yields a net positive benefit starting from batch=1, the smallest batch size we tested.** — SOLO is net positive at every batch size we tested. At batch=1, SOLO achieves 5,504 vs default's 2,882 TPS on OpenPayments (+91%), and 9,008 vs 4,133 on Food (+118%). The one-time planning overhead (17.5s on OpenPayments, see W2) is recouped early in the inference process. At batch=128, gains grow to +1,410% (OpenPayments) and +205% (Food). Detailed results across all batch sizes are provided in our response to Dkyu-Q3.
>
>
> **W5 (Over Traditional DB ordering).** We compared 12 methods including NDV/cardinality sort (Lemire & Kaser, 2011), frequency-based column ordering, and clustering-based reordering. SOLO outperforms all on every dataset: on OpenPayments, SOLO 75.93 vs NDV 72.47 vs GGR 62.72; on Food (200 cols), SOLO 154.91 vs NDV 140.30 vs GGR 99.58. Full 12-method ranking across four datasets in our response to Sx1V-Q2.

---

> > ### Author Rebuttal · Reviewer_UyFA · 2026-04-03
> >
> > I appreciate the authors’ effort in the rebuttal. I will raise my rating.

---

### Official Review · Reviewer_qjXL · 2026-03-11

**Soundness:** 3
**Presentation:** 3
**Significance:** 3
**Originality:** 3
**Overall Recommendation:** 4
**Confidence:** 4

**Summary:**

This paper addresses the critical performance bottleneck of the prefill phase in LLM-augmented database analytics. By establishing an isomorphism between prefix KV-cache reuse and radix tree topology, the authors propose SOLO, a prefix-cache-aware layout optimization algorithm that reorders rows and columns to maximize prefix overlap. SOLO leverages a greedy column ordering strategy with a tight 1/(M−1) worst-case approximation guarantee and optimal row ordering via lexicographical sorting, achieving up to 90.3% prefill throughput improvement and 242× reduction in planning overhead compared to state-of-the-art baselines. The work integrates seamlessly into existing database pipelines and validates its effectiveness across multiple real-world datasets and semantic operators (filtering, join, top-k), while preserving task accuracy.

**Compliance With Llm Reviewing Policy:**

Affirmed.

**Final Justification:**

The authors have addressed my concerns. I am keeping the positive score.

**Key Questions For Authors:**

see W1-W5

**Limitations:**

yes

**Strengths And Weaknesses:**

S1: Establishes a rigorous theoretical framework, including formal proofs of the isomorphism between prefix KV-cache reuse and radix tree topology, the NP-hardness of column ordering, and the greedy algorithm’s tight 1/(M−1) worst-case approximation guarantee, laying a solid foundation for the proposed method.

S2: Features comprehensive experimental validation to fully demonstrate SOLO’s effectiveness across semantic operators (filtering, join, top-k).

S3: Optimizes the algorithm’s time complexity to O(M2N+MNlogN) via hash-based counting and incremental prefix state maintenance, eliminating repeated sorting and ensuring practical scalability for moderate-scale tables.

S4：Enables non-intrusive integration into existing database pipelines (e.g., LOTUS) without altering core system designs, supporting proxy-oracle LLM cascades and various semantic operators, while preserving task accuracy (variation ≤1%), making it highly applicable to real-world scenarios.

S5: Introduces a novel insight of linking prefix cache reuse to radix tree topology, shifting layout optimization from heuristic shuffling to a well-defined tree-shaping problem, distinguishing it from prior methods (e.g., GGR, OPHR) lacking formal theoretical guarantees.

W1: Exclusively adopts a "one-row-per-request" processing model without comparing it with batch processing, nor justifying why single-row processing is superior.

W2: Fails to explore concurrent request initiation with prefix-sharing constraints, which is a standard feature in practical LLM serving systems (e.g., vLLM), despite the flexible row order enabling such parallelism.

W3: Lacks explicit space complexity analysis, including no formal theoretical bounds or empirical memory overhead data for large-scale tables (e.g., N=1e7,M=200).

W4: Requires full-table scanning for initialization, which is infeasible for ultra-large tables (e.g., N>1e7) and misaligned with real-world "one-by-one" request patterns.

W5: The algorithm still has room for potential improvement, e.g., hybrid DP-Greedy strategies (local DP for small column subsets + global Greedy) or probabilistic data structures (e.g., HyperLogLog++), which could further balance optimization performance and efficiency.

---

> ### Author Rebuttal · Authors · 2026-03-31
>
> # Response to Reviewer qjXL
>
> **W1 (One-row-per-request).** We first justify the choice, then compare with multi-row-per-request. One-row-per-request is the standard model for semantic database operators (e.g., LOTUS, Patel et al. 2025), where each row requires an independent LLM judgment. To compare, we packed 1/2/4/8 rows into a single prompt on OpenPayments using Qwen2.5-7B (semantic filter: "Is this recipient in California?"), measuring accuracy and per-row latency. Per-row latency increases from 204 ms to 307 ms (+50%), and accuracy fluctuates between 98.6%–99.1%, slightly below the 100% of one-row. multi-row-per-request is slower with no accuracy benefit.
>
> | Rows/Prompt | Accuracy | Time/Row |
> |---|---|---|
> | 1 | 100.0% | 204 ms |
> | 2 | 98.6% | 249 ms |
> | 4 | 99.1% | 262 ms |
> | 8 | 99.0% | 307 ms |
>
>
> **W2 (Concurrent request initiation with prefix-sharing). We have explored concurrent request initiation with prefix-sharing constraints; SOLO's row ordering inherently enables prefix-sharing across concurrent requests.** Full details are in our response to Dkyu-Q3. Gains grow with concurrency: +91% at batch=1 to +1,410% at batch=128 on OpenPayments, and +118% to +205% on Food.
>
> **W3 (Space complexity). SOLO's planner requires O(MN) space and scales to N=10^7 within 8 GB.**
>
> *Theoretical bound.* Algorithm 1 maintains three structures: the encoded attribute matrix R_enc (O(MN)), the rolling prefix state F (O(N)), and candidate composite keys K_cand (O(N)). Total space is O(MN), dominated by R_enc.
>
> *Empirical validation.* We profiled peak memory on Food (M=200, the widest table):
>
> | N (M=200) | Peak Memory |
> |---|:---:|
> | 10^5 | 0.32 GB |
> | 10^6 | 0.88 GB |
> | 5*10^6 | 3.88 GB |
> | 10^7 | 7.67 GB |
>
> | M (N=10^7) | Peak Memory |
> |---|:---:|
> | 50 | 2.10 GB |
> | 100 | 3.96 GB |
> | 150 | 5.83 GB |
> | 200 | 7.69 GB |
>
> Peak memory scales linearly with both N and M, confirming O(MN).
>
> **W4 (Full-table scanning for ultra-large tables). SOLO scales well to ultra-large tables: even at N=10^7 rows and M=200 columns, planning completes in ~180s on a single machine.**
>
> *Feasibility for ultra-large tables.* We ran the full pipeline (column ordering + lexicographic row sorting) on Food (M=200, the widest table). Larger versions (10^6–10^7) were generated by sampling with replacement from the original 100K rows:
>
> | N (M=200) | Planning Time |
> |---|:---:|
> | 10^5 | 30s |
> | 10^6 | 36s |
> | 5*10^6 | 86s |
> | 10^7 | 180s |
>
> | M (N=10^7) | Planning Time |
> |---|:---:|
> | 20 | 24s |
> | 50 | 51s |
> | 100 | 105s |
> | 150 | 143s |
> | 200 | 183s |
>
> *Applicability under one-by-one request patterns.* SOLO is applicable in both primary LLM-augmented database analytics and concurrent serving scenarios. In the primary setting of LLM-augmented database analytics (Patel et al., 2025), the full table is available before query execution begins, making full-table scanning directly applicable. In concurrent serving scenarios — where a user submits many single-row requests simultaneously to the LLM — vLLM's continuous batching naturally collects these concurrent requests into a single forward pass; SOLO can reorder rows within each such batch to maximize prefix sharing among the relational prompts, without requiring access to the full table. In either case, the planning cost is well-amortized: on OpenPayments, 17.5s of total overhead (planning + reorder) enables 6 hours of prefill savings (see UyFA-W2).
>
> **W5 (Algorithm improvements). Both suggestions are valuable.** Hybrid DP-Greedy can tighten the approximation for wide tables by solving small column subsets exactly before global greedy assembly; HyperLogLog++ can reduce per-iteration cost by replacing exact group counts with approximate cardinality estimates. We will include both options in the future work section of the revised paper.

---

> > ### Author Rebuttal · Reviewer_qjXL · 2026-04-06
> >
> > Thanks for the rebuttal. I am keeping the positive score.

---

### Official Review · Reviewer_Dkyu · 2026-03-11

**Soundness:** 3
**Presentation:** 3
**Significance:** 3
**Originality:** 3
**Overall Recommendation:** 5
**Confidence:** 3

**Summary:**

This paper studies how to reorder rows and columns of relational data so that prompts sent to an LLM share longer prefixes, enabling more prefix KV-cache reuse during prefill. The main idea is to view a fixed column order as inducing a radix tree over tuples, prove that lexicographic row sorting is optimal for that fixed tree, and then greedily choose a column ordering to reduce tree size. The resulting system, SOLO, is integrated into a semantic analytics pipeline and evaluated on four datasets and several operator types, with reported gains in prefill throughput and much lower planning overhead than prior reordering baselines.

**Compliance With Llm Reviewing Policy:**

Affirmed.

**Final Justification:**

I have raised my score based on the rebuttal comments and discussion

**Key Questions For Authors:**

- In your experimental vLLM setup, can a request reuse prefixes from any still-cached prior request, or only from the immediately preceding one in practice? If nonadjacent reuse is possible, why is Equation (5) the right optimization target, and can you provide evidence that it correlates with realized hit rate or throughput?

- For Proposition 3.5, do you intend the claim to hold under generic column independence, or only under the product-uniform model proved in the appendix? If the latter, will you revise the main text accordingly?

- How sensitive are the gains in Figure 5 and the planning-overhead comparison in Figure 6 to KV-cache budget, batching policy, and prompt length regime? A brief response with even a small ablation could materially change my assessment.

- Can you provide a clearer fairness story for ACLR/GCLR/GGR, ideally under matched planning-time budgets, and include main-paper quality metrics for the operator workloads? If those results hold up, my evaluation would improve.

**Limitations:**

yes

**Strengths And Weaknesses:**

Strengths

-The core abstraction is strong and useful. The radix-tree formulation is not just decorative. It gives a concrete way to reason about prefix sharing, and Figure 3 is one of the paper’s most effective figures because it visually explains why column ordering and row ordering play different roles.

- The paper translates theory into a practical planner. Algorithm 1 is simple enough to deploy, and the implementation choice of maintaining rolling prefix-state IDs is a good systems detail rather than a hand-wavy description.

- The empirical story is internally consistent. I appreciated that Figure 5 does not only show throughput. In Figure 5(b-d), the paper walks through prefix overlap, reused KV tokens, and hit rate. This makes the mechanism behind Figure 5(a) much more credible than a bare speedup plot would.

-The complexity comparison is easy to understand and matters. Table 1 is concise and useful: compared with the exponential OPHR and quadratic-recursive GGR style costs, the claimed (\mathcal{O}(M^2N + MN\log N)) planner is appealing for realistic table sizes.

-The systems integration angle is relevant. Figure 4 helps show that the method is not an isolated optimizer but a drop-in planner for an existing semantic-operator pipeline.


Weaknesses

- The optimization objective appears mismatched to the backend semantics used in practice. On Page 4, Assumption 3.1 says only the prefix states from the immediately previous prompt are guaranteed reusable. But modern prefix-caching systems often allow reuse from older cached prefixes as long as they remain resident. This is not a minor modeling shortcut. It goes straight to the heart of the method: SOLO optimizes adjacent overlap under Equation (5), while the deployed system may realize reuse in a more global way. The paper needs either a convincing justification that adjacent-overlap optimization is a good surrogate under the actual vLLM setup, or an experimental study showing strong correlation between its objective and realized cache reuse.

- Experimental robustness is underexplored. The gains in Figure 5(a) are substantial, especially on OpenPayments and Food, but they are shown for one main cache-budget regime and one main serving stack. Since the title and abstract pitch this as a generally useful optimization for LLM-augmented database analytics, I expected more sensitivity analysis. The absence of cache-budget sweeps is particularly noticeable because the abstract explicitly conditions gains on a fixed prefix-cache budget.

- The baseline comparison is not fully fair or fully informative. Table 2 lists good baseline families, but the main paper does not give enough detail on the computational budgets assigned to ACLR and GCLR. If stochastic search is budget-limited, then both quality and runtime depend heavily on that budget. A more convincing evaluation would show either matched-time comparisons or a Pareto curve of planning overhead versus achieved prefix reuse / throughput.

- A main-paper ablation of row-only versus column-only versus full SOLO is missing. The method is explicitly two-part, and the paper’s own logic depends on the interaction between row scheduling and column serialization. The appendix reportedly has such an ablation, and I think at least a compact version belongs in the main paper.

---

> ### Author Rebuttal · Authors · 2026-03-31
>
> # Response to Reviewer Dkyu
>
> We thank Reviewer Dkyu for the thorough and insightful review.
>
>
> **Q1/W1 (Adjacent reuse vs global cache — Assumption 3.1). (1) Can requests reuse prefixes from any still-cached prior request? Yes, a request can reuse prefixes from any still-cached prior request, not only the preceding one.** vLLM's global prefix cache supports this natively, and we do not restrict this behavior in our setup.
>
> (2)**Why is adjacent overlap (Eq. 5) well-justified when non-adjacent reuse is available?** Intuitively, compared with non-adjacent requests, the just-processed preceding request's prefixes are much more likely to remain cache-resident, while older entries face eviction under finite budgets. Adjacent overlap thus measures the prefill tokens each request can most reliably skip.
>
> Empirically, for each dataset we ran 7 ordering methods and measured actual reused KV tokens and prefill TPS under vLLM's full global cache (any-request reuse enabled). Pearson r between adjacent overlap and each metric:
>
> | Dataset | Cols | overlap ↔ reused KV (r) | overlap ↔ prefill TPS (r) |
> |---|---|---|---|
> | DMV | 20 | 0.997 | 0.995 |
> | OpenPayments | 90 | 0.993 | 0.998 |
> | GoogleAnalytics | 54 | 1.000 | 0.998 |
> | Food | 200 | 0.999 | 0.981 |
>
> All r > 0.98 across datasets ranging from 20 to 200 columns, confirming that adjacent overlap is a tight surrogate for realized cache hit rate and throughput.
>
> **Q2 (Proposition 3.5 scope).** Proposition 3.5 is established under the product-uniform model (Theorem C.6 in the appendix). We will revise the context to state this assumption explicitly.
>
> **Q3 / W2 (Sensitivity to KV-cache budget, batching policy, and prompt length). SOLO's throughput gains (Figure 5) are robust across all three dimensions, and planning overhead (Figure 6) is independent of all of them.** Planning cost depends only on data scale — O(M²N + MN log N) in the number of columns M and rows N; cache budget, batch size, and prompt length are serving-stage parameters.
>
> **(1) KV-cache budget** (batch size = 8, prompt length = 50 tokens fixed).
>
> SOLO's hit rate gain is stable across an 18× range of cache sizes (variation < 2%):
>
> | KV Cache Size | OpenPayments | Food |
> |---|---|---|
> | 0.3 GB | +69.5% | +60.8% |
> | 0.5 GB | +69.4% | +61.0% |
> | 2.0 GB | +69.2% | +61.2% |
> | 3.5 GB | +68.9% | +61.5% |
> | 5.5 GB | +68.5% | +61.7% |
>
> **(2) Batch size / batching policy** (cache budget = 0.5 GB, prompt length = 50 tokens fixed).
>
> SOLO's advantage persists at all batch sizes and grows with concurrency (prefill TPS, tokens/s):
>
> | Batch | OpenPayments Default | OpenPayments SOLO | Food Default | Food SOLO |
> |---|---|---|---|---|
> | 1 | 2,882 | 5,504 | 4,133 | 9,008 |
> | 8 | 5,555 | 30,188 | 4,939 | 13,951 |
> | 32 | 5,937 | 57,064 | 5,089 | 15,202 |
> | 128 | 6,074 | 91,691 | 5,102 | 15,559 |
>
> Higher concurrency amplifies the benefit: gains grow from +91% (batch=1) to +1,410% (batch=128) on OpenPayments, and +118% to +205% on Food.
>
> **(3) Prompt length** (cache budget = 0.5 GB, batch size = 8 fixed).
>
> We varied system prompt length P while keeping data ordering fixed — SOLO's hit rate gain (%):
>
> | P (tokens) | Description | OpenPayments | Food |
> |---|---|---|---|
> | 5 | minimal | +71.7 | +61.9 |
> | 50 | short task | +69.4 | +61.0 |
> | 200 | few-shot | +62.5 | +58.1 |
> | 500 | long few-shot | +52.3 | +53.2 |
>
> As P grows, hit rate gain (%) decreases because data tokens become a smaller fraction of total tokens, but the absolute number of additionally reused tokens is unchanged.
>
> **Q4 / W3 (Baseline fairness — ACLR/GCLR/GGR).**
>
> **We conducted a matched planning-time comparison under identical hardware and a unified time budget to provide fair computational budget.** All methods ran on an Intel Xeon Silver 4210R CPU with 40 cores and 504 GB RAM with a time ceiling of 187.5 s (GGR's full deterministic runtime on OpenPayments), giving stochastic methods the maximum possible budget. SOLO completes in 1.5 s.
>
> The time-quality Pareto comparison below reports average prefix overlap (shared prefix attributes) at each elapsed time. Even after 187.5 s (125× SOLO's planning time), ACLR achieves 72% of SOLO's overlap, GCLR 49%, and GGR 83%.
>
> | Method | 1.5s | 10s | 30s | 187.5s |
> |---|---|---|---|---|
> | SOLO | 75.93 | — | — | — |
> | ACLR | 6.27 | 26.04 | 37.49 | 54.97 |
> | GCLR | 26.11 | 26.11 | 25.78 | 36.85 |
> | GGR | — | — | — | 62.72 |
>
> **W4 (Main-paper additions).**
>
> We will add two items to the revised paper:
>
> - **Ablation.** A compact version of the row-only vs. column-only vs. full SOLO ablation (Appendix D.2, Figure 8) will be moved to the main paper.
> - **Quality preservation.** SOLO does not materially alter task accuracy. Across all datasets and reordering strategies, worst-case |ΔAcc| < 1.2% and Stability > 97% (Appendix I, Table 8). We will add a compact summary to the main paper.

---

> > ### Author Rebuttal · Reviewer_Dkyu · 2026-03-31
> >
> > N/A

---

### Official Review · Reviewer_Sx1V · 2026-03-13

**Soundness:** 3
**Presentation:** 3
**Significance:** 3
**Originality:** 3
**Overall Recommendation:** 5
**Confidence:** 3

**Summary:**

This paper studies prefix cache reuse in LLM-augmented relational database analytics. In relational settings, prefix reuse is often limited by the default prompt construction strategy, where tuples are processed in row-wise request order and attributes are serialized in column-wise in-prompt order. Such a layout fails to create shared prefixes between adjacent LLM invocations, thereby limiting prefix cache effectiveness.The authors observe that prefix cache reuse in relational workloads is structurally equivalent to path sharing on a radix tree induced by column priority. Based on this insight, they propose Solo, a fast and robust prefix-cache-aware scheduling framework that reorders both (i) the per-tuple LLM invocation sequence and (ii) the within-prompt field serialization order to maximize prefix overlap between adjacent requests.

[Score increased after authors addressing W1 and W2].

**Compliance With Llm Reviewing Policy:**

Affirmed.

**Ethical Review Concerns:**

n.a.

**Final Justification:**

[Score increased after authors addressing W1 and W2].

**Key Questions For Authors:**

Q1. Please provide performance under this baseline and quantify the improvement of Solo over the unoptimized system on four datasets (see W1).

Q2. Please compare Solo against simple reordering heuristics (e.g., random order, single-key sorting, or basic lexicographic sorting or other representatives) (see W2).

**Limitations:**

Address W1 and W2.

**Strengths And Weaknesses:**

S1. The paper establishes a rigorous theoretical framework for prefix-cache-aware relational prompt layout optimization in LLM-augmented database analytics, revealing an isomorphism between prefix cache reuse and radix tree topology.

S2. The authors prove that for any fixed column order (where the column-ordering problem itself is NP-hard proved by authors), lexicographical row sorting is optimal for maximizing prefix reuse.

S3. The authors design an efficient approximation algorithm for column ordering with a tight worst-case approximation guarantee of 1/(M-1), providing both theoretical justification and practical applicability.

W1. The paper misses a default baseline without any optimization in the main paper, i.e., running the LLM-augmented analytics pipeline using the original data order and the default prompt layout. Reporting this baseline is essential to quantify the absolute effectiveness and efficiency of the unoptimized system, and to clearly measure the gains brought by optimization techniques (e.g., Solo and GGR).

W2. The evaluation also lacks simple and intuitive data-reordering baselines. Besides the proposed methods, the authors should include at least a few lightweight heuristics, such as:
- Random order: a sanity-check baseline to show the sensitivity to ordering.
- Single-key sorting: sort tuples by one selected attribute (e.g., the most frequent / highest-cardinality column, or the first attribute in the prompt) to create partial prefix locality.
- Frequency-based column priority + lexicographic sort: choose a column order based on global token frequency (or distinct count), then lexicographically sort rows accordingly (a simple heuristic that does not require sophisticated planning).
- Clustering-based reorder: group tuples by similarity (e.g., using hashing or simple string prefix grouping on a key field) and process groups consecutively to increase prefix overlap.

---

> ### Author Rebuttal · Authors · 2026-03-31
>
> #  Response to Reviewer Sx1V
>
> We thank Reviewer Sx1V for the constructive suggestions that helped us strengthen the evaluation.
>
> **Q1 (Default baseline and performance gains).** **We report the absolute effectiveness and efficiency of the unoptimized system and quantify the gains brought by SOLO on all four datasets**, using two metrics: overlap (average shared leading attributes, κ in Eq. 2) and prefix cache hit rate measured end-to-end under vLLM:
>
> | Dataset | #Cols | Default Overlap | SOLO Overlap | Default Hit Rate | SOLO Hit Rate |
> |---------|:---:|:---:|:---:|:---:|:---:|
> | DMV | 20 | 0.80 | **12.72** | 1.8% | **35.2%** |
> | OpenPayments | 90 | 10.49 | **75.93** | 23.7% | **76.3%** |
> | GoogleAnalytics | 54 | 1.16 | **44.68** | 3.2% | **62.1%** |
> | Food | 200 | 0.00 | **154.91** | 0.0% | **71.1%** |
>
> SOLO delivers substantial hit rate gains over the unoptimized default: +33.4% (DMV), +52.6% (OpenPayments), +58.9% (GoogleAnalytics), and +71.1% (Food).
>
> **Q2 (Simple baselines).** **We compared SOLO against all requested heuristics, including random order, single-key sorting, frequency-based column priority, and clustering-based reorder.** Below is the complete 12-method comparison on OpenPayments (90 columns):
>
> | Rank | Method | Category | Overlap | Hit Rate |
> |:---:|--------|----------|:---:|:---:|
> | 1 | **SOLO** | **Ours** | **75.93** | **76.3%** |
> | 2 | freq_lex | Heuristic: freq col + lex sort | 72.48 | 66.1% |
> | 3 | NDV sort | Compression: cardinality order | 72.47 | 66.1% |
> | 4 | GGR | Prior art: group recursion | 62.72 | 63.9% |
> | 5 | col_only | Ablation: SOLO col ordering only | 54.47 | 65.3% |
> | 6 | cluster_prefix | Heuristic: prefix grouping | 15.94 | 34.2% |
> | 7 | row_only | Ablation: lex sort, default cols | 14.44 | 29.9% |
> | 8 | single_key (min NDV) | Heuristic: sort by lowest-cardinality col | 11.32 | 25.8% |
> | 9 | single_key (max freq) | Heuristic: sort by most-frequent col | 11.32 | 25.8% |
> | 10 | single_key (1st col) | Heuristic: sort by first attribute | 10.51 | 23.8% |
> | 11 | default_raw | No optimization | 10.49 | 23.7% |
> | 12 | random_row | Sanity check | 3.36 | 7.3% |
>
> SOLO achieves the highest overlap and hit rate. Key comparisons on OpenPayments:
> - vs GGR (prior SOTA): SOLO 75.93 vs GGR 62.72 (+21% overlap)
> - vs best heuristic (freq_lex): SOLO 75.93 vs 72.48 (+4.8%)
> - vs single-key sorts: SOLO 75.93 vs best single-key 11.32 — simple sorting captures little of the available prefix structure
> - Random (3.36) < default (10.49), confirming that ordering quality matters
>
> The same ranking holds across DMV, GoogleAnalytics, and Food. On DMV (20 cols):
>
> | Rank | Method | Overlap | Hit Rate |
> |:---:|--------|:---:|:---:|
> | 1 | **SOLO** | **12.72** | **35.2%** |
> | 2 | GGR | 11.65 | 30.9% |
> | 3 | freq_lex | 11.16 | 28.8% |
> | 4 | NDV | 11.16 | 28.8% |
> | 5 | col_only | 8.26 | 31.4% |
> | 6 | cluster_prefix | 3.00 | 8.6% |
> | 7 | single_key (min NDV) | 1.83 | 6.1% |
> | 8 | single_key (max freq) | 1.80 | 6.1% |
> | 9 | row_only | 1.00 | 1.8% |
> | 10 | single_key (1st col) | 1.00 | 1.8% |
> | 11 | default | 0.80 | 1.8% |
> | 12 | random | 0.54 | 0.04% |
>
> On GoogleAnalytics (54 cols):
>
> | Rank | Method | Overlap | Hit Rate |
> |:---:|--------|:---:|:---:|
> | 1 | **SOLO** | **44.68** | **62.1%** |
> | 2 | freq_lex | 40.68 | 56.8% |
> | 3 | NDV | 40.68 | 56.8% |
> | 4 | GGR | 39.51 | 56.9% |
> | 5 | col_only | 27.02 | 52.5% |
> | 6 | cluster_prefix | 4.06 | 7.4% |
> | 7 | single_key (min NDV) | 2.16 | 6.4% |
> | 8 | single_key (max freq) | 2.16 | 6.4% |
> | 9 | row_only | 2.06 | 4.0% |
> | 10 | single_key (1st col) | 2.00 | 3.5% |
> | 11 | default | 1.16 | 3.2% |
> | 12 | random | 0.37 | 1.8% |
>
> On Food (200 cols, widest table):
>
> | Rank | Method | Overlap | Hit Rate |
> |:---:|--------|:---:|:---:|
> | 1 | **SOLO** | **154.91** | **71.1%** |
> | 2 | freq_lex | 140.43 | 66.5% |
> | 3 | NDV | 140.30 | 66.3% |
> | 4 | col_only | 127.49 | 66.7% |
> | 5 | GGR | 99.58 | 47.7% |
> | 6 | cluster_prefix | 2.00 | 0.0% |
> | 7 | single_key (max freq) | 1.00 | 0.0% |
> | 8 | single_key (min NDV) | 1.00 | 0.0% |
> | 9 | single_key (1st col) | 0.00 | 0.0% |
> | 10 | row_only | 0.00 | 0.0% |
> | 11 | default | 0.00 | 0.0% |
> | 12 | random | 0.00 | 0.0% |
>
> SOLO consistently outperforms all counterparts across all datasets. The margin over the best heuristic is larger on wider tables, specifically, DMV 20 cols: +1.6 attributes; Food 200 cols: +14.6 attributes, where the column ordering problem is harder.

---

> > ### Author Rebuttal · Reviewer_Sx1V · 2026-04-01
> >
> > All of my questions were answered.

---

### Decision · Program_Chairs · 2026-04-30

**Decision:**

Accept (regular)

**Comment:**

There was broad consensus among the reviewers about the importance of studying prefix cache reuse in LLM-augmented relational database analytics. The solution of the paper was satisfactory to the reviewers and there was broad consensus on accept, which is my recommendation. I urge the authors to incorporate the extra baselines and experiments performed in the rebuttal stage in the final version of the paper, as it addressed some weaknesses of the reviewers.